



# Exploring the capabilities of electrical resistivity tomography to study subsea permafrost

Mauricio Arboleda-Zapata[1], Michael Angelopoulos[2], Pier Paul Overduin[2], Guido Grosse[2], Benjamin M. Jones[3], and Jens Tronicke[1]

[1]University of Potsdam, Institute of Geosciences, Potsdam, Germany
[2]Alfred Wegener Institute, Helmholtz Centre for Polar and Marine Research, Potsdam, Germany
[3]University of Alaska Fairbanks, Institute of Northern Engineering, Fairbanks, AK, United States of America

**Correspondence:** Mauricio Arboleda-Zapata (mauricio.arboleda@geo.uni-potsdam.de)

**Abstract.** Sea level rise and coastal erosion have inundated large areas of Arctic permafrost. Submergence by warmer and saline waters increases the rate of inundated permafrost thaw compared to sub-aerial thawing on land. Studying the contact between the unfrozen and frozen sediments below the seabed, also known as the ice-bearing permafrost table (IBPT), provides valuable information to understand the evolution of sub-aquatic permafrost, which is key to improving and understanding coastal erosion

prediction models and potential greenhouse gas emissions. In this study, we use data from 2D electrical resistivity tomography (ERT) collected in the nearshore coastal zone of two Arctic regions that differ in their environmental conditions (e.g., seawater depth and resistivity) to image and study the subsea permafrost. The inversion of 2D ERT data sets is commonly performed using deterministic approaches that favor smoothed solutions, which are typically interpreted using a user-specified resistivity threshold to identify the IBPT position. In contrast, to target the IBPT position directly during inversion, we use a layer-

based model parameterization and a global optimization approach to invert our ERT data. This approach results in ensembles of layered 2D model solutions, which we use to identify the IBPT and estimate the resistivity of the unfrozen and frozen sediments, including estimates of uncertainties. Additionally, we globally invert 1D synthetic resistivity data and perform sensitivity analyses to study, in a simpler way, the correlations and influences of our model parameters. The set of methods provided in this study may help to further exploit ERT data collected in such permafrost environments as well as for the design

of future field experiments.

## 1 Introduction

In arctic coastal regions, contemporary subsea permafrost thawing starts following the inundation caused by sea level rise and coastal erosion. Seawater is typically warmer than mean annual air temperatures, and the presence of saltwater (mostly through diffusive processes) lowers the freezing point of the seabed (Harrison and Osterkamp, 1978; Are, 2003). Additionally,

groundwater flow of freshwater from inland areas might play an important role in thawing permafrost (Frederick and Buffett, 2015; Pedrazas et al., 2020), comparable to warm discharge from large rivers (Shakhova et al., 2017). Subsea permafrost is estimated to contain a large quantity of organic carbon (Sayedi et al., 2020), which can decompose microbially to generate carbon dioxide and/or methane after the permafrost thaws. Furthermore, gas hydrates are present in subsea permafrost and



may act as an additional source of methane if they dissociate (Ruppel and Kessler, 2017). Understanding the development of

permafrost degradation rates helps to fine-tune predictive models of greenhouse gas emissions that may represent a positive

feedback for climate warming (Schuur et al., 2015). Furthermore, the correlation between permafrost degradation and coastal

erosion proposed by Are (2003) and Overduin et al. (2012, 2016) might be used to refine coastal dynamics models.

Subsea permafrost is a perennially cryotic (< 0 °C) layer (or body) of sediments underneath a marine water column (An-

gelopoulos et al., 2020a). These sediments can be frozen or unfrozen. A layer of unfrozen ground in a permafrost area is known

as talik, and in particular, the perennially cryotic unfrozen sediments (forming part of the permafrost) are known as cryopegs

(Permafrost Subcommittee, 1988). Cryopegs can be isolated pockets or layers and are commonly found along Arctic coasts in

saline marine sediments that are exposed offshore following a marine regression, for example, due to isostatic uplift (O'Neill

et al., 2020). Offshore, cryotic unfrozen sediment in between the water column and frozen ground is still generally referred

to as a talik (Osterkamp, 2001). The subsea permafrost that contains ice is known as ice-bearing permafrost, and when the

soil particles are cemented together by ice, it is termed ice-bonded permafrost (Permafrost Subcommittee, 1988). Because

traditional geophysical methods (e.g., ERT and seismic techniques) can only distinguish between sediments without or low ice

content from those with high ice content (note that direct sampling is required for a quantitative estimation of ice-content), we

refer to them as unfrozen and frozen sediments in this study, and the interface that separates them is the ice-bearing permafrost

table (IBPT). Imaging and determining the position of the IBPT (e.g., using geophysical or borehole data) is important for a

better process understanding of subsea permafrost evolution and to infer degradation rates. For example, dividing the depth to

the IBPT by the time since inundation results in the mean annual degradation rate (e.g., Are, 2003; Overduin et al., 2012, 2016).

Among the most used geophysical imaging techniques to study the subsea permafrost are different electromagnetic and

seismic methods as well as electrical resistivity tomography (ERT) (Scott et al., 1990; Kneisel et al., 2008; Hubbard et al.,

2013). Electromagnetic induction (EMI) methods are promising techniques to map both the top and bottom boundaries of

the permafrost and might be used for a wide range of water depths (e.g., Sherman et al., 2017). EMI methods can properly

work under conductive seawater layers, while the use of ground-penetrating radar (GPR) is limited to freshwater bottom-fast

ice environments characterized by high electrical resistivity values as found in delta areas (e.g., Stevens et al., 2009). Seismic

methods have been employed widely in deep-water environments (e.g., Rekant et al., 2015; Brothers et al., 2016) and more

recently, researchers have used recordings of ambient seismic noise in shallow waters to map the IBPT (e.g., Overduin et al.,

2015a). The ERT method (which is the focus of this study) is a suitable tool to investigate the resistivity distribution of the

unfrozen sediments (e.g., talik and cryopeg) and for studying and delineating the IBPT position (e.g., Sellmann et al., 1989;

Overduin et al., 2012, 2016; Angelopoulos et al., 2019, 2020b; Angelopoulos, 2022; Pedrazas et al., 2020).


In marine ERT surveying, floating electrodes are typically used to inject a current and measure potential differences that are

used to calculate apparent electrical resistivity data. The measured values (in the summer season of the Arctic) are influenced by

the resistivity and thickness of the water layer and by the unfrozen and frozen sediments. The ERT method can detect the IBPT



but does not necessarily distinguish non-cryotic from cryotic taliks above the IBPT. The resistivity of seawater depends mainly
on the amount of dissolved salts (which is affected by water inflows from rivers and the cycles of sea ice melting, freezing, and
brine release) and temperature, and is commonly in the range of 0.1 Ωm to 40 Ωm (e.g., Sellmann et al., 1989; Lantuit et al.,
2011). On the other hand, the resistivity of the underlying sediments is influenced by porosity, pore size, grain size, water and
ice content, porewater salinity, and temperature (Kneisel et al., 2008; Wu et al., 2017). For example, the resistivity of unfrozen
sediments typically ranges from 1 Ωm to 25 Ωm (e.g., Sellmann et al., 1989; Overduin et al., 2012; Angelopoulos et al., 2019),
while the resistivities of frozen sediments might vary from 10 Ωm up to more than 1,000 Ωm (e.g., Overduin et al., 2012, 2016;
Pedrazas et al., 2020; Rangel et al., 2021). The higher the ice content, the less conductive is the medium (Pearson et al., 1986;
Fortier et al., 1994; Kang and Lee, 2015). In cases where the resistivity of the frozen sediments is several orders of magnitude
higher than the resistivity of the overlying unfrozen sediments, the electrical current injected through the electrodes is expected
to be channeled through the more conductive layers (e.g., Spitzer, 1998) resulting in a limited penetration of the current system
into the frozen sediment layers.

When analyzing ERT data collected in subsea permafrost environments, defining an appropriate inversion and model pa-
rameterization strategy is critical for deriving reliable resistivity models and interpreting these models in terms of the IBPT
position. For example, when a priori information suggests that the nature of the contact between the unfrozen and frozen
sediments is gradual, a grid-based model parameterization and a local inversion algorithm favoring vertical and/or horizontal
smoothness in the final models might be an appropriate choice (e.g., Loke and Barker, 1996; Günther et al., 2006). Here, the
experience of the interpreter might help to guess a specific resistivity threshold value to define the IBPT position (e.g., Overduin
et al., 2016; Sherman et al., 2017; Angelopoulos et al., 2021). Additionally, one may also consider different gradient-based
image processing approaches to extract interfaces from the inverted resistivity model (e.g., Hsu et al., 2010; Chambers et al.,
2012). Finally, when we have ample evidence of a sharp contact between the unfrozen and frozen sediments (e.g., Overduin
et al., 2015b; Angelopoulos et al., 2020a), a layer-based model parameterization combined with a local and/or global inversion
algorithms might be more suitable (e.g., Auken and Christiansen, 2004; Akça and Basokur, 2010; De Pasquale et al., 2019;
Arboleda-Zapata et al., 2022).

In this study, we adapt the inversion and ensemble interpretation strategies as proposed by Arboleda-Zapata et al. (2022)
to study submarine permafrost environments of the Arctic in terms of the resistivity distribution of the unfrozen and frozen
sediments including estimates of uncertainties, also around the IBPT. We analyze and compare ERT data collected at two
field sites in the Arctic characterized by different environmental conditions (e.g., regarding seawater depth and resistivity,
coastal erosion rate, and the sediments porewater chemistry), which have been recorded using different acquisition settings
(e.g., different electrode spacing and spatial sampling). Additionally, we generate and interpret ensembles of globally inverted
1D electrical data to get a deeper understanding of the inverse problem for typical resistivity distributions in these kinds of
environments. Finally, we also implement local and global sensitivity analysis to recognize the most influential parameters
during 2D and 1D inversions.



## 2 Study sites

Our field data have been collected at two field sites; one offshore of the southern part of the Bykovsky Peninsula in the Siberian Laptev Sea (Fig. 1a), and the other one offshore of Drew Point in the Alaskan Beaufort Sea (Fig. 1e). To relate these data sets to the site-specific environmental settings, we summarize the main characteristics of each study area in a regional framework.

### 2.1 Regional setting of Bykovsky Peninsula

The Bykovsky Peninsula is located in northeastern Siberia in the vicinity of the Lena River Delta. The peninsula is mainly
characterized by the presence of ice-rich sediments (volumetric ice content exceeding 80 %, also known as the Yedoma Ice Complex) that accumulated during the Late Pleistocene (Schirrmeister et al., 2002; Grosse et al., 2007). The Yedoma deposits extend to 15 m below sea level (Grigoriev, 2008). The sediments at or below sea level are composed of silt, sand, and gravel with variable grain size distributions (Grosse et al., 2007). During the early to middle Holocene, a general landscape transformation started resulting in a thermokarst-dominated relief characterized by thermo-erosional valleys and thermokarst lakes
(Schirrmeister et al., 2002; Grosse et al., 2007). The coastal erosion rates at the peninsula range between 0.4 m per year to 1.5 m per year (maximum values of up to 10 m per year) mainly caused by storms and thermomechanical erosion of ice-rich sediments (Lantuit et al., 2011). The seawater around the peninsula is strongly influenced by freshwater and sediments originating from the Lena River (Lantuit et al., 2011). Additionally, the resistivity of the seawater is influenced by seasonal sea ice freezing and melting as shown by Lantuit et al. (2011) who report resistivity values for the eastern shore of the Bykovsky Peninsula
of less than 1 $\Omega$m in winter and above 10 $\Omega$m in summer. Similar water resistivity values are also reported by Overduin et al. (2016) for the seawater near Muostakh Island. The depth of the seawater for the southern part of the Bykovsky peninsula deepens 2 m within a distance of 100 m from the shoreline and increases to 5 m at about 2,000 m from the coast (Lantuit et al., 2011; Fuchs et al., 2021).

### 2.2 Regional setting of Drew Point

Drew Point is located on the coast of the Alaskan Beaufort Sea. The local geology is characterized by glaciomarine, fine-grained, ice-rich sediments deposited in the late Pleistocene (Black, 1964; Ping et al., 2011). The inland geomorphology is characterized by coastal bluffs (typically between 3 − 5 m high), thermokarst channels and lakes, and ice-wedge polygons on tundra plains (maximum elevation of ∼ 10 m) (Barnhart et al., 2014; Jones et al., 2018). The average coastal erosion
rate between 1979 and 2002 was around 9 m per year (Jones et al., 2009) and increased for the period 2002 to 2016 up to approximately 20 m per year (Jones et al., 2018). Lück (2020) reports brackish water conductivities observed during fieldwork in July 2018 of 0.4 − 0.5 $\Omega$m, with weak stratification visible in depth profiles. The depth of the seawater offshore of Drew Point deepens 2 m within a distance of 500 m from the shoreline and increases to 3 m at distances about 2,000 m from the coast (Jones et al., 2018).



**Figure 1.** Location and ERT data of our field studies. a) Bykovsky field site located at the coast of the Laptev Sea in Northern Siberia (Sakha Republic, Russian Federation) and e) Drew Point field site located at the coast of the Beaufort Sea in Northern Alaska (AK, United States of America), where the read lines indicate ERT profile locations and the black dash line in e) indicates the position of the 1955 coastline for Drew Point (Jones et al., 2008). b) The recorded bathymetric profile along the ERT profile for Bykovsky and f) for Drew Point indicating the 1969 (Schirrmeister et al., 2018) and 1955 coastline positions, respectively. The current coast position for both profiles is at $x \approx -10$ m. c)-d) 2D and 1D representation of the recorded raw ERT data for Bykovsky, and g)-h) for Drew Point (see text for details). Satellite image Bykovsky: Worldview3 satellite product from September 2[nd], 2016; copyright Digital Globe. Satellite image Drew Point: Planet satellite image from September 3[rd], 2017.



## 3 Data acquisition

In marine ERT data acquisition, there is typically an excellent coupling between the floating electrodes and the seawater. This allows us to perform voltage measurements while the boat pulling the electrode streamer is in motion (preferably at constant speed) and, thus, to efficiently measure also profiles with a length in the order of km. The sources of errors during data acquisition are mainly related to misalignments of the electrode streamer (e.g., due to water currents), the precision of electrode positioning (which are given relative to boat position), vertical oscillation of electrodes (e.g., due to wavy conditions), and surface area limitation of injection voltage. Furthermore, due to the large variety of environmental settings, one must tailor the survey parameters to each field site, which includes varying the electrode spacing, the transmitter voltage, the measurement duration, the boat speed, the digital resolution of the potential measurements, and the sampling frequency.

In Table 1, we compare the acquisition parameters for our ERT data from Bykovsky and Drew Point, which were collected during two fieldwork campaigns in July 2017 and July 2018, respectively. The two ERT data sets were collected using an Iris$^{TM}$ Syscal Pro Deep Marine system employing a streamer cable with 13 equally spaced floating electrodes. The resistivity measurements were acquired using a reciprocal Wenner-Schlumberger array configuration, where current was injected through the inner pair of electrodes and quasi-symmetric voltages were measured simultaneously with 10 channels using the outer pair of electrodes (e.g., Overduin et al., 2012). The transmitter voltage was set at approximately 48 V at Bykovsky while, at Drew Point, it was reduced to 24 V to avoid exceeding the electrode surface area limits. Additionally, different electrode spacings were used. The Bykovsky data were recorded using a 10 m spacing between electrodes while, at Drew Point, 5 m spacing was chosen because the rapid coastal erosion rates suggested that the IBPT position at this field site should be shallower than at our Siberian field site for a given distance offshore. To collect the data along every profile, a cable was towed behind a small inflatable boat and voltages were measured as the boat moved at approximately constant speed ($\sim 1.1$ m/s) perpendicular to the shore. The Bykovsky soundings were collected at a spacing of five to seven meters (540 measurements in total, 418 m long) and the Drew Point soundings at one to two meters ($1,830$ measurements in total, 854 m long). The electrode positions were estimated relative to the position of a GPS aboard the boat (assuming a straight streamer cable). Complementary to the ERT measurements, we also recorded the water depth at each sounding location using a Garmin echo sounder attached to the boat (Fig. 1b, and f). Furthermore, we measured water conductivity and temperature at different depths (the mean values are shown in the last two rows of Table 1) using a Sontek$^{TM}$ CastAway device (also known as CTD). In general, at our Drew Point field site, the seawater was shallower, more conductive, and slightly cooler than at our Bykovsky field site.

The measured apparent resistivities $\rho_a$ are presented as pseudosections in Fig. 1c and g. Here the $x$ coordinates represent the center position of each quadripole, and the vertical axis (levels) represent the relative penetration; i.e., level $= 0$ is the shortest quadripole, while level $= 9$ is the quadripole with maximum electrode spacing. The range of $\rho_a$ for Bykovsky is 5.9 $\Omega$m to 45 $\Omega$m and for Drew Point 0.9 $\Omega$m to 6.3 $\Omega$m. The lower $\rho_a$ values at Drew Point are mainly due to the higher conductivity of the seawater at the Alaskan coast, which is less influenced by freshwater discharge from large rivers than at our Bykovsky field site.

**Table 1.** Acquisition parameters for our ERT data sets and further site-specific information for our two field sites.

|  | Bykovsky, Siberia | Drew Point, Alaska |
| --- | --- | --- |
| Number of electrodes | 13 | 13 |
| Electrode spacing (m) | 10 | 5 |
| Transmitter voltage (V) | 48 | 24 |
| Sounding separation (m) | 5 to 7 | 1 to 2 |
| Length of profile (m) | 418 | 854 |
| Number of data points | 540 | 1830 |
| Water resistivity ($\Omega m$) | 13.7 | 0.5 |
| Water temperature (°C) | 7 | 5.5 |

Additionally, we found it useful to plot the data as 1D sounding curves (Fig. 1d and h) because this allows for additional visual
inspections of data quality. For example, we noticed for our Bykovsky data that levels seven and eight are characterized by
higher noise levels than all other levels. In contrast, the Drew Point data not show obvious variations in data quality depending
on the level number (i.e., along the 1D sounding curves).

## 4   Methodology

In this study, we follow the workflow of Arboleda-Zapata et al. (2022) who propose a layer-based model parameterization to
globally invert 2D ERT data, which is used to generate an ensemble of representative model solutions. For completeness, we
present a brief summary of this workflow in the following. For a more detailed analysis, we will also address complementary
strategies such as 1D inversion tests as well as local and global sensitivity analyses.

### 4.1   2D layer-based model parameterization

One of the most important steps in any geophysical inversion workflow is defining a model parameterization that can prop-
erly represent the studied geological environment. Because a priori information suggests a layered subsurface (i.e., unfrozen
sediments overlying frozen sediments) at both of our field sites, we choose a layer-based model parameterization considering
an unstructured mesh with local refinements along the interfaces separating individual layers. Additionally, because resistivity
variations within each layer are negligible compared to the variations between different layers, we assume homogeneous layers;
i.e., each layer may be characterized by one resistivity value. For more complex geological settings, one might allow for lateral
and/or vertical variation within the layers (e.g., Auken and Christiansen, 2004; Akça and Basokur, 2010). To parameterize the
interface geometry that defines the contact between the individual layers, we may use different functions based, for example,
on spline interpolation (e.g., Koren et al., 1991), Fourier coefficients (e.g., Roy et al., 2021), or sums of arctangent functions





(Gebrande, 1976). For this study, we adopt a strategy based on the sum of arctangent functions because it allows for abrupt
and smooth changes along the interfaces (e.g., Roy et al., 2005; Rumpf and Tronicke, 2015). Allowing for abrupt changes is
considered to be important in permafrost environments where high structural variability is often found; for example, due to
inundated thermokarst structures (Angelopoulos et al., 2021), pingo-like features, bottom-fast ice versus floating ice regime
transitions in winter, or changes in the ratio of coastal erosion vs. degradation rate; i.e., changing from a period of fast thawing
and low coastal erosion to a period of fast coastal erosion and slow thawing can result in a heterogeneous structure of the IBPT
(e.g., Overduin et al., 2016).

## 4.2   Inversion strategy

During inversion, we search for a combination of model parameters (i.e., those describing the geometry of interfaces and the
resistivities of the homogeneous layers) that minimizes the root mean squared logarithmic error (RMSLE). Because we aim
to find an inverse model independent of a reference or starting model, we use a global inversion strategy based on the particle
swarm optimization (PSO) technique, which was originally introduced by Kennedy and Eberhart (1995). Over the last decade,
the PSO algorithm has been widely used to invert different types of geophysical data sets because it has proven to be an ef-
fective tool for finding different local minima in objective functions with complicated topography (e.g., Tronicke et al., 2012;
Fernández-Martínez et al., 2017).


In a first step, the PSO requires defining a set of particles (where each particle represents a different model) that are initialized
with random parameters (bounded within realistic physical ranges). This defines our model space. In the following iterations,
the position of each particle is updated considering the best global position found so far by the entire swarm (i.e., the particle
with the best fit performance in terms of the RMSLE), the best local position (i.e., the best fit performance in the history
of each particle), and the inertia (i.e., the direction in which the particle is moving). These parameters are weighted and
perturbed with random numbers drawn from a uniform distribution which helps avoid getting trapped in a local minimum.
For every particle and every iteration, we calculate the forward response in a different finite-element model mesh using the
python library pyGIMLi (Rücker et al., 2017). This allows for local refinements around the interfaces and, thus, to calculate the
forward response with high precision and with a reasonable amount of time (Arboleda-Zapata et al., 2022). At the end, when
the optimization reaches the maximum number of iterations or a minimum threshold in the objective function, we save the final
best model. Using different seeds of the random number generator, we repeat this process until we obtain an ensemble $M_{F0}$
consisting of several hundred independent models and an ensemble of corresponding residuals $\delta_{F0}$. In this study, each residual
vector is calculated as the difference between the observed and the corresponding modeled log-apparent resistivity values.

## 4.3   Ensemble interpretation

In a first step, to ease our ensemble analysis and interpretation in a pixel-wise fashion, all models in $M_{F0}$ are interpolated using
the nearest-neighbor algorithm on a densely discretized structured mesh (note that we use a unstructured mesh during inversion,





Sect. 4.1). In a second step, we perform a cluster analysis using the $k$-means algorithm (MacQueen, 1967) and considering $\boldsymbol{M}_{F0}$ and $\boldsymbol{\delta}_{F0}$ as input to group similar solutions from our ensembles. To find an optimal number of clusters $n_k$, we use the criterion proposed by Caliński and Harabasz (1974) supported by a visual inspection of the clustering results. Finally, we characterize in a pixel-wise fashion each found cluster $\boldsymbol{M}_{Fi}$ and $\boldsymbol{\delta}_{Fi}$ (where $i = 0, 1, ..., n_k$, note $i = 0$ correspond to the whole ensemble and $i > 0$ to the clustered ensembles) by the median values $\mu_{1/2}(\boldsymbol{M}_{Fi})$ and $\mu_{1/2}(\boldsymbol{\delta}_{Fi})$ and the interquartile ranges IQR$(\boldsymbol{M}_{Fi})$ and IQR$(\boldsymbol{\delta}_{Fi})$. Additionally, we describe $\boldsymbol{\delta}_{Fi}$ in an overall fashion assessing the RMSLE$(\boldsymbol{\delta}_{Fi})$, the IQR$(\boldsymbol{\delta}_{Fi})$, and the quantile 90 % $q_{90}(\boldsymbol{\delta}_{Fi})$.

### 4.4 1D inversion

Often, we prefer 2D inversion algorithms in comparison to 1D strategies; especially for field data where the subsurface situation and its complexity are largely unknown. However, to investigate and understand, for example, the relationship between specific model parameters and the influence of a priori information and constraints, 1D approaches (also considering synthetic data examples) represent helpful interpretation tools (e.g., Sen and Stoffa, 1996; Malinverno, 2002).

In this study, we use 1D models consisting of five parameters, the depth of the seawater $z_w$, the depth of the contact between frozen and unfrozen sediments $z_{pt}$ (i.e., IBPT), the water resistivity $\rho_w$, the resistivity of the unfrozen sediments $\rho_{uf}$, and the resistivity of the ice-bearing permafrost $\rho_p$. As for our 2D examples, we also consider PSO to invert our 1D synthetic data. Because for such 1D inversions the computational cost is significantly lower than 2D problems, we can run several tests and create larger model ensembles. We use such a 1D approach to tackle some specific questions regarding the considered application. For example, we investigate how constraining the depth of the water layer and its resistivity affects the final ensemble of 1D model solutions. Additionally, the limited number of parameters in our 1D model parameterization strategy allows us to study in a simpler way the posterior correlation matrix as proposed by Sen and Stoffa (2013). Although in this study we not consider cluster analysis to classify our 1D ensembles (as implemented for our 2D analyses), this step may be adapted in future studies (e.g., investigating more complex model scenarios).

### 4.5 Sensitivity analysis

Sensitivity analysis is a powerful tool that can provide additional information to improve system or process understanding (Wainwright et al., 2014). In the context of subsea permafrost applications, several studies have shown the potential of the ERT method to image the IBPT position (e.g., Sellmann et al., 1989; Overduin et al., 2012). However, the sensitivity distribution of the ERT model parameters for such environments characterized by high resistivity contrasts (up to several orders of magnitude) between frozen and unfrozen sediments is poorly understood. Adding sensitivity analysis to the interpretation workflow helps investigate the impact of our chosen model parameterization and the used constraints. Furthermore, such sensitivity studies might also help optimize ERT acquisition geometries and strategies before a field campaign.





In this study, we use 2D-local and 1D-global sensitivity analyses. To investigate which regions of the 2D discretized model have the greatest influence on our objective function, we consider the difference-based local sensitivity method of Günther et al. (2006), which is available within the pyGIMLi library (Rücker et al., 2017). For example, we assess the sensitivity of the shortest electrode configurations to understand if the corresponding measurements are influenced by both the water layer and the underlying unfrozen sediments. In turn, this helps to evaluate the reliability of imaging the uppermost water layer (e.g., for

measurements where no CTD measurements are available). Furthermore, the longest electrode spreads (corresponding to the deepest levels in 2D pseudosections) and/or cumulative sensitivity distributions provide information on whether our ERT data are sensitive to the IBPT and/or the frozen sediments. For 1D model parameterizations and synthetic studies (considering $z_w$, $z_{pt}$, $\rho_w$, $\rho_{uf}$, and $\rho_p$ as described in Sect. 4.4), we use the variance-based global sensitivity method of Sobol (Sobol, 2001; Saltelli et al., 2008) as implemented in the python library SALib (Herman and Usher, 2017). Using this approach, we aim to

understand how the total influence of the considered parameters might be affected by variations in $\rho_p$ and $z_{pt}$.

## 5 Results

In the following, we present the 2D inversion results for the Bykovsky and Drew Point data sets in two separate subsections. Each subsection is complemented with 1D inversion results of synthetic data simulated considering the site-specific environmental and electrode settings as well as with a 2D-local and a 1D-global sensitivity analysis.

### 5.1 Bykovsky

The geological settings of the Bykovsky area are described in Sect. 2.1 and a summary of the acquisition parameters are provided in Table 1. We invert the $540$ apparent resistivity measurements recorded along a $418$ m long profile (Fig. 1c) using a layer-based model parameterization as described in Sect. 4.1 and a PSO-based inversion strategy as outlined in Sect. 4.2. We parameterize our model with two interfaces (considering five nodes for each interface), one for the seabed and the other for the

IBPT. In total, this results in 36 model parameters, which describe the structure of the interfaces and the layer resistivities. In the PSO, we use $60$ particles and a maximum of $600$ iterations as stopping criterion. To obtain a single inverted model (i.e., after one inversion run), we have to evaluate the forward response $36,000$ times, which takes on average $\sim 40$ hours on a single core of a modern desktop computer. We repeat these inversion runs considering different initial seeds of the random number generator (note that this approach allows for a straightforward parallelization when multiple cores are available) until we obtain

an ensemble $\boldsymbol{M}_{F0}$ consisting of 690 models.

### 5.1.1 Ensemble analysis

After the inversion, we interpolated all models to a refined structured mesh before performing any posterior statistical analyses (see Sect. 4.3). In Fig. 2a and b, we show the $\mu_{1/2}(\boldsymbol{M}_{F0})$ and IQR($\boldsymbol{M}_{F0}$) models calculated from the Bykovsky model

ensemble. The $\mu_{1/2}(\boldsymbol{M}_{F0})$ model indicates that $\rho_{uf}$ is $\sim 4$ $\Omega$m and $\rho_p$ is $\sim 60,000$ $\Omega$m. However, when analyzing individual



models, we note a bimodal distribution of $\rho_p$ (some models with $\rho_p < 2,000$ Ωm and others with $\rho_p > 100,000$ Ωm) which is also illustrated by increased IQR($M_{F0}$) values for the lowermost layer. These observations already indicates different groups of models with distinct resistivity characteristics.

In the next step, we performed cluster analysis (Sect. 4.3) and found that our ensemble $M_{F0}$ can be divided into two model families ($M_{F1}$ and $M_{F2}$). In Fig. 2b-c and e-f, we present the $\mu_{1/2}(M_{Fi})$ and IQR($M_{Fi}$) models (where $i = 1, 2$). Comparing these models illustrates that $M_{F1}$ and $M_{F2}$ show a similar IBPT shape dipping toward the open sea (i.e., depth of the IBPT increases with increasing profile distances). However, the IBPT position in $M_{F1}$ is shallower than in $M_{F2}$. We learn from this that if $\rho_p$ increases, the depth of the IBPT increases (resulting in thicker unfrozen sediments), also near the coast.

According to the depth of the IBPT and its gradients in profile direction, we laterally subdivide the models into three main parts. The first part is found at $x < 130$ m and is characterized by a gentle dipping slope with minor convexities and concavities. The second part is found at $130 < x < 280$ m, where the IBPT is relatively flat with a minor change in depth at $x \sim 200$ m. Finally, the abrupt change at $x = 280$ m marks the transition to the third part, which is characterized by a rather deep IBPT (with depths $> 20$ m) and extends until the end of the profile.


    We assess the fit performance in a pixel-wise and in an overall fashion for the residuals associated to the ensemble containing all models $\delta_{F0}$, as well as for the two clustered model families $\delta_{F1}$ and $\delta_{F2}$ (Fig. 3). Thus, we calculate $\mu_{1/2}(\delta_{Fi})$ and IQR($\delta_{Fi}$) (where $i = 0, 1, 2$) in a pixel-wise fashion and present them as pseudosections in Fig. 3a-f. When comparing these pseudosections to each other, we notice that the $\mu_{1/2}(\delta_{Fi})$ and IQR($\delta_{Fi}$) indicate similar fits of the data in terms of

amplitudes and pseudosection patterns. The abrupt change from positive to negative residuals at $x \simeq 200$ m coincides with the highest point of the bathymetric profile for $x > 150$ m which also corresponds to a general change in the gradients of the bathymetric profile (Fig. 1b). Therefore, a 3D subsurface structure (which cannot be explained by our 2D inversion strategy) and related 3D effects are a reasonable explanation of the discussed features in the residuals. For example, because the landscape was partly covered by lakes (a heat source) prior to seawater submergence, lateral temperature gradients and heterogeneous

sediment properties could affect subsurface resistivity and its 3D variations. The overall statistics RMSLE($\delta_{Fi}$), IQR($\delta_{Fi}$), and $q_{90}(\delta_{Fi})$ (where $i = 0, 1, 2$) are presented as histograms in Fig. 3g-i. The histograms are characterized by bimodal distributions, especially evident in all shown RMSLE($\delta_{Fi}$) histograms. When comparing the histograms of $\delta_{F1}$ and $\delta_{F2}$, we notice that they follow similar distributions (although in $\delta_{F1}$ there are less models). From these analyses of the residuals, we are not able to prefer one of the model families and, thus, we perform some synthetic exercises to deepen our understanding of this

inverse problem and the found model solutions.

### 5.1.2   1D inversion of synthetic data

To complement our understanding of the formulated inverse problem, we perform 1D inversions of a synthetic data set created considering a 1D subsurface model (see "Input model" in Table 2) as described in Sect. 4.4. The 1D subsurface model pa-


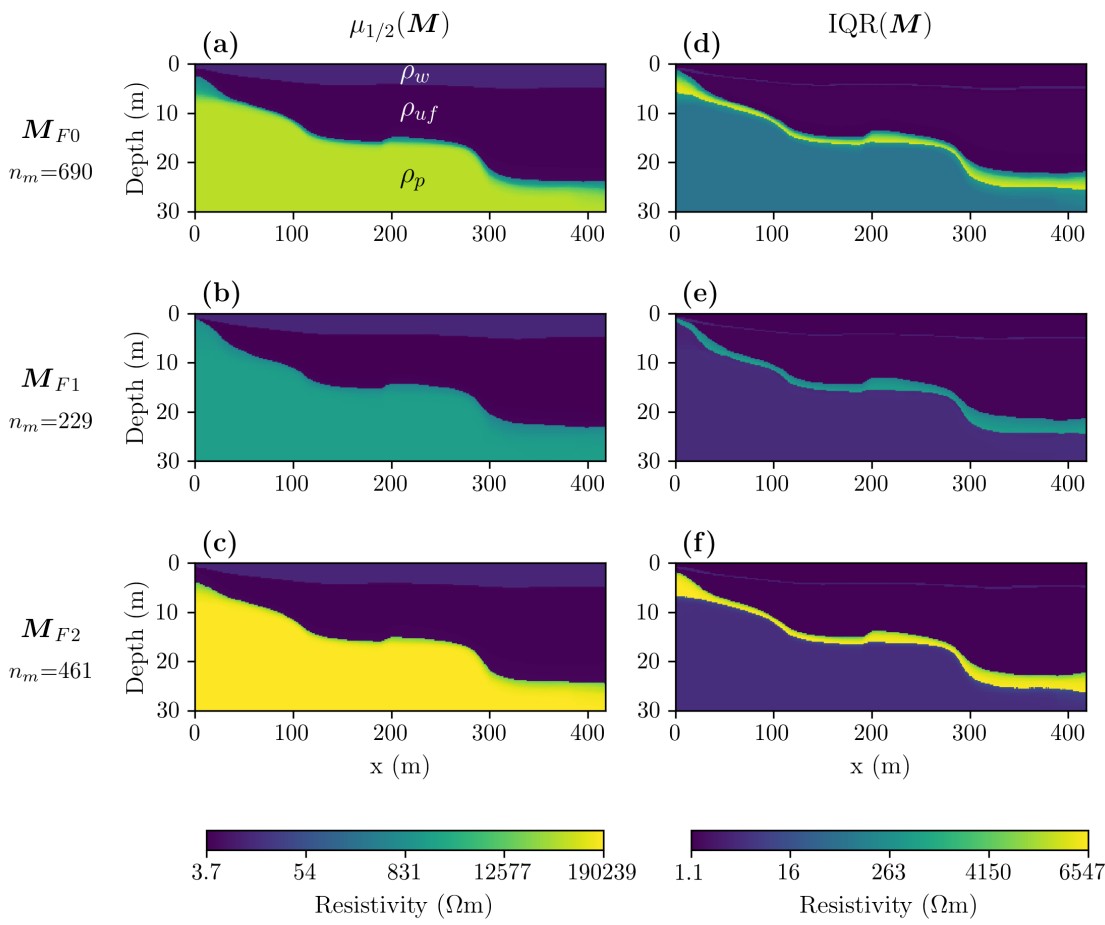

**Figure 2.** Inversion results for the Bykovsky data set illustrated as summary statistics for all obtained models $\boldsymbol{M}_{F0}$ and for two model families $\boldsymbol{M}_{F1}$ and $\boldsymbol{M}_{F2}$ as found by cluster analysis. a)-c) Median and d)-f) interquartile range models. For each $\boldsymbol{M}_{Fi}$, $n_m$ represents the number of models in the corresponding ensemble.

rameters were chosen by analyzing our 2D model solutions (e.g., Fig. 2b-c at $x \simeq 150$ m). We calculate the forward response of 10 quadripoles (which is equivalent to one sounding curve in Fig. 1d) considering the same electrode configurations as used for recording the Bykovsky field data (Table 1). We invert the simulated apparent resistivity data using two scenarios for constraining $z_w$ and $\rho_w$, while the constraints for all other parameters remain unchanged (see Table 2). The resulting inverted models are shown in Fig. 4a and c. For all models, we have achieved RMSLE $< 0.028$, which is equivalent to the noise level

applied to the calculated synthetic data and comparable to the RMSLE achieved for the 2D inversion results of the Bykovsky field data. Comparing the results shown in Fig. 4a and c illustrates that constraining the water layer significantly decreases the non-uniqueness of the inverse problem. We also notice that the median model represents a good approximation to the input



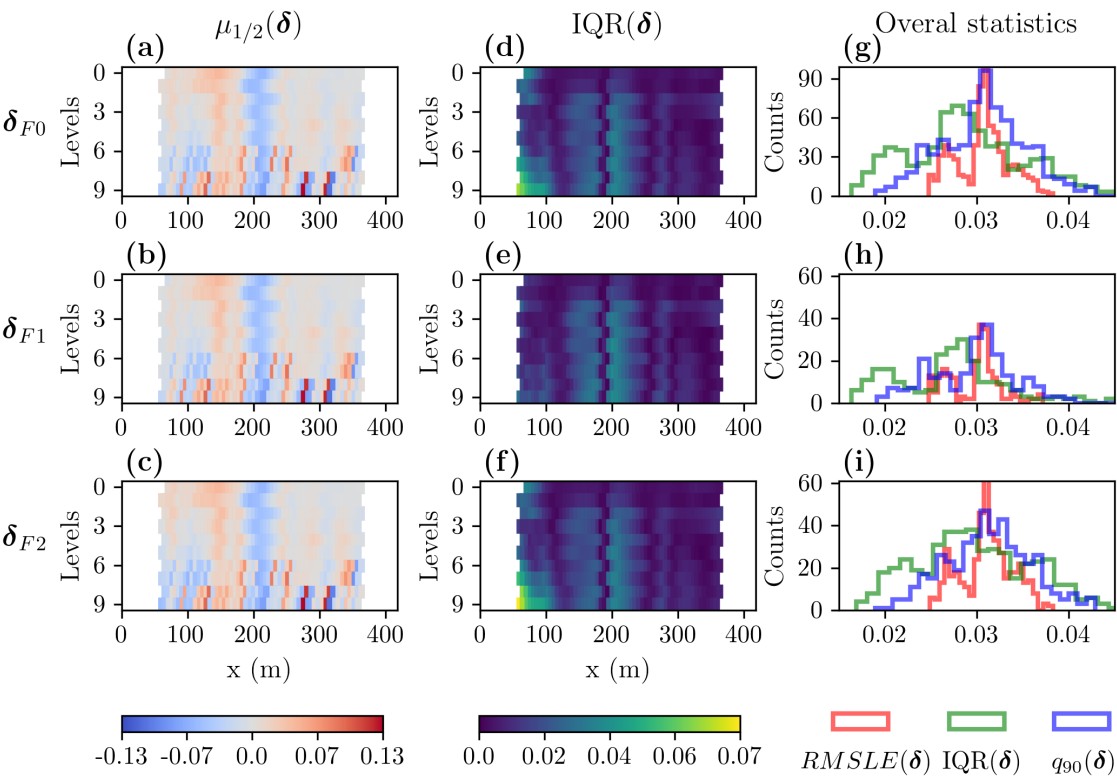

**Figure 3.** Summary statistics of the residuals for the Bykovsky data set corresponding to all models $\delta_{F0}$ and for the two clustered families $\delta_{F1}$ and $\delta_{F2}$. a)-c) Median and d)-f) interquartile range calculated in a pixel-wise fashion. g)-i) Histograms illustrating the overall distribution of different statistical measures including RMSLE($\delta$), IQR($\delta$), and $q_{90}(\delta)$.

model except for $\rho_p$ which is overestimated because the quantile 25 % and 75 % models ($q_{25}$, $q_{75}$) indicate resistivities larger than the input $\rho_p$. Additionally, from all models visualized in Fig. 4a and c, we calculate the corresponding posterior correlation

matrices (Fig. 4b and d). For both cases, we see that $[z_{pt}, \rho_{uf}]$ is the model parameter pair with the highest positive correlation while the rest of the model parameter pairs are characterized by negative correlation with different amplitudes (except for pairs with $\rho_p$ which show correlations approaching zero).

### 5.1.3 Sensitivity analysis

To understand the sensitivity distribution for our three-layer model (representing seawater and unfrozen sediments overlying frozen sediments), we calculate the cumulative sensitivity, and the sensitivity for the shortest and widest quadripoles considering two model scenarios (Fig. 5). In the first scenario, we consider the same input model as for the 1D inversion exercise (Table 2). In the second scenario, we set $z_{pt} = 25$ m while all other parameters remain unchanged. From the cumulative sensitivity





**Table 2.** Parameters of the 1D synthetic model of Bykovsky and for two scenarios indicating the lower and upper bounds parameter constraints.

|  | Input model | Scenario 1 | Scenario 2 |
| --- | --- | --- | --- |
| Depth seawater $z_w$ (m) | 4.5 | 3, 6 | 4, 5 |
| Depth IBPT $z_{pt}$ (m) | 15 | 6.5, 25 | 6.5, 25 |
| Resistivity seawater $\rho_w$ (Ωm) | 13.7 | 1, 50 | 11, 15 |
| Resistivity unfrozen sediments $\rho_{uf}$ (Ωm) | 4 | 1, 100 | 1, 100 |
| Resistivity permafrost $\rho_p$ (Ωm) | 4,000 | 1, 200,000 | 1, 200,000 |

plots (Fig. 5a and d), we learn that areas of sensitivities extend throughout the layer of unfrozen sediments for both scenarios.

This suggests that we can interpret our inverted models even underneath the outer electrode positions; i.e., if the boat together with the electrode streamer is moving toward the right (i.e., increased $x$ coordinates) to collect additional sounding curves, our interpretation of the inverted model should start at $x \sim -60$ m (a more conservative model interpretation might start at $x \sim -25$ m). When analyzing Fig. 5b and e, we see that the shortest quadripole is sensitive to both the water layer and the unfrozen sediments. For a wider electrode spacing and an IBPT located at a depth of 15 m (Fig. 5c), the sensitivities are focused

around the inner electrodes but also with some minor contributions from the outer electrodes (note the reddish colors in the unfrozen sediments at $x < -60$ m and at $x > 60$ m), which may be critical when significant 2D or 3D resistivity variations are present. For a deeper IBPT (Fig. 5f), we notice that we are still sensitive at depths of $\sim 25$ m; however, the lateral extensions of the sensitivity patterns within the unfrozen sediments appear to be reduced.

As noticed in our 2D sensitivity analysis, the high resistivity contrast between the unfrozen and frozen sediments seems to limit the penetration depth up to the IBPT. To complement and better understand our results of 2D sensitivity analysis, we investigate the global sensitivities (Sect. 4.5) of different 1D model parameterizations. Specifically, we use models where $z_w = 4.5$ m, $\rho_w = 13.7$ Ωm, and $\rho_{uf} = 4$ Ωm are fixed, while $\rho_p$ varies between 10 Ωm and 10,000 Ωm (eight values in total) and the IBPT is located at three different depths; i.e., $z_{pt} = 25$ m, $z_{pt} = 15$ m, and $z_{pt} = 5$ m (Fig. 6a-c). Note, defining eight

different values for $\rho_p$ and three for $z_{pt}$ results in 24 different 1D models. For the calculation of the total sensitivity for each of our five parameters in these 24 models, we set the parameters ranges to $z_w = [4, 6]$ m, $z_{pt} = [6.5, 30]$ m, $\rho_w = [0.2, 20]$ Ωm, $\rho_{uf} = [1, 20]$ Ωm, and $\rho_p = [5, 20000]$ Ωm. For these specific models and parameter ranges, our results (Fig. 6) suggest $\rho_w$ is the most influential parameter followed by $\rho_{uf}$, which shows approximately half of the influence compared to $\rho_w$. The influence of $z_{pt}$ is slightly larger than $z_w$, although $z_{pt}$ is set up with a higher range than $z_w$. Furthermore, although we allow

$\rho_p$ to vary over three orders of magnitude the result of this sensitivity analysis demonstrates that $\rho_p$ is the parameter with the lowest influence, but it is not null as indicated by the results of our 2D sensitivity analyses (Fig. 5a and d). Interestingly, we also notice in Fig. 6a-c that in general when increasing $\rho_p$ (for $\rho_p < 100$ Ωm) the total sensitivity index of the other parameters tend to decrease.


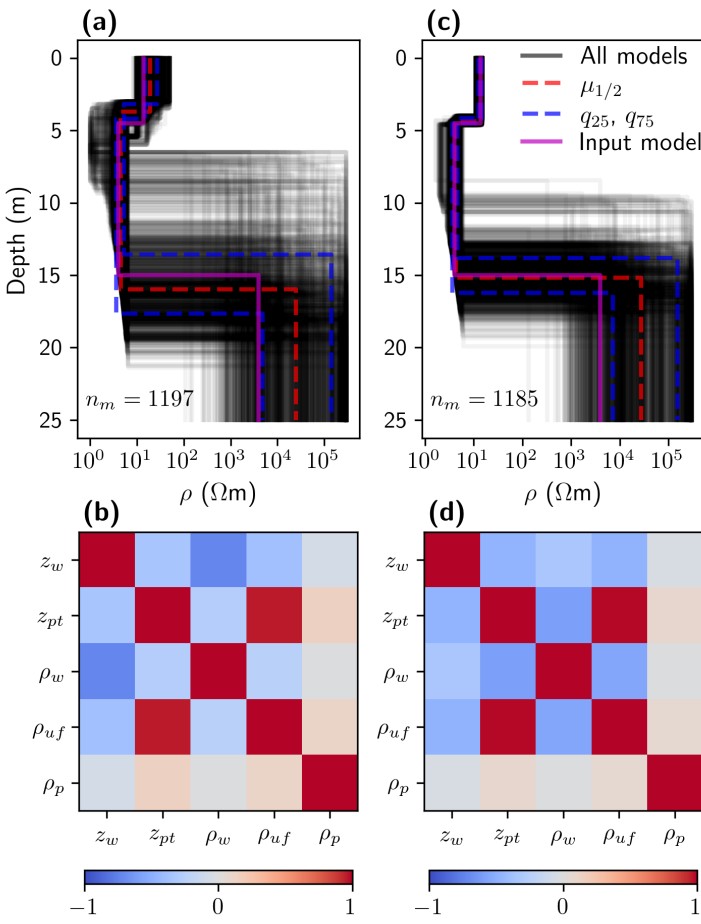

**Figure 4.** 1D inversion results of synthetic data for 1D subsurface scenarios developed for the Bykovsky field site. a) Ensemble of solutions and b) the corresponding correlation matrix for scenario 1 (water layer parameters with large freedom during inversion), and c)-d) the same for scenario 2 (with constrained $z_w$ and $\rho_w$). Note, black lines in a) and c) are plotted with transparency and, therefore, the darker areas indicate higher densities model. Each ensemble contains $n_m$ model solutions.

## 5.2 Drew point

The geological settings of the Drew Point area are described in Sect. 2.2 and a summary of the acquisition parameters is provided in Table 1. We invert the 1830 apparent resistivity measurements recorded along an 854 m long profile (Fig. 1g) considering a layer-based model parameterization consisting of two interfaces and five nodes for each interface (see Sect. 4.1). In total, this results in 36 model parameters, which describe the structure of the interfaces and the layer resistivities. In the PSO, we use 30 particles and a maximum of 400 iterations as stopping criterion. To obtain a single inverted model, we have to

evaluate the forward response 12,000 times, which takes on average 57 hours on a single core of a modern desktop computer.

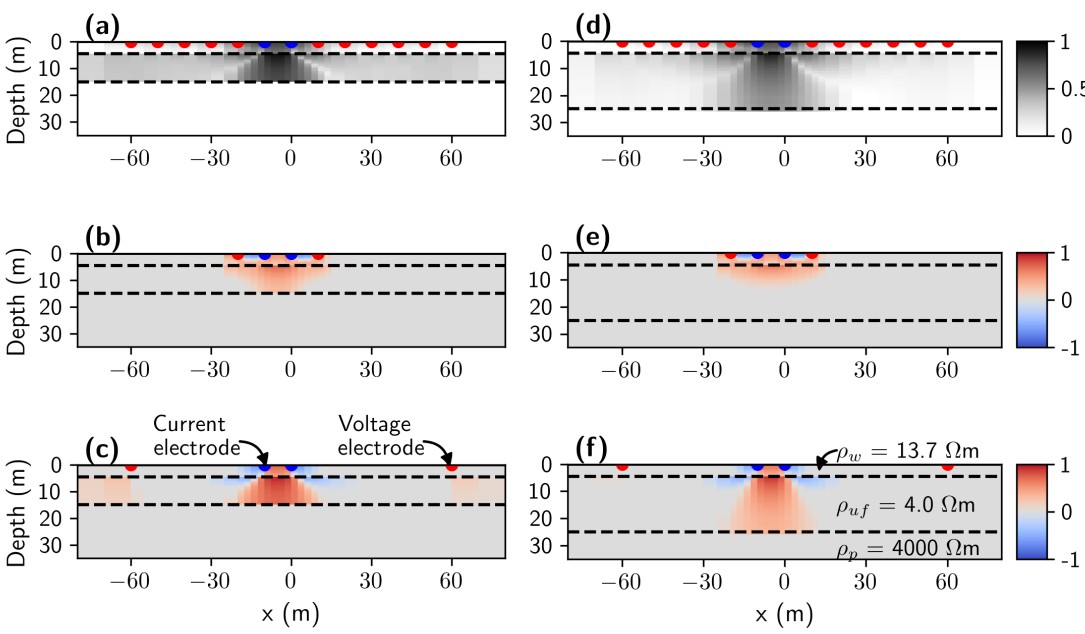

**Figure 5.** 2D normalized sensitivities for two different model scenarios developed for the Bykovsky field site. Position of the the IBPT at a depth of a)-c) 15 m, and d)-f) 25 m. From top to bottom, we show the cumulative sensitivity and the sensitivity for the shortest and widest quadripole, respectively.

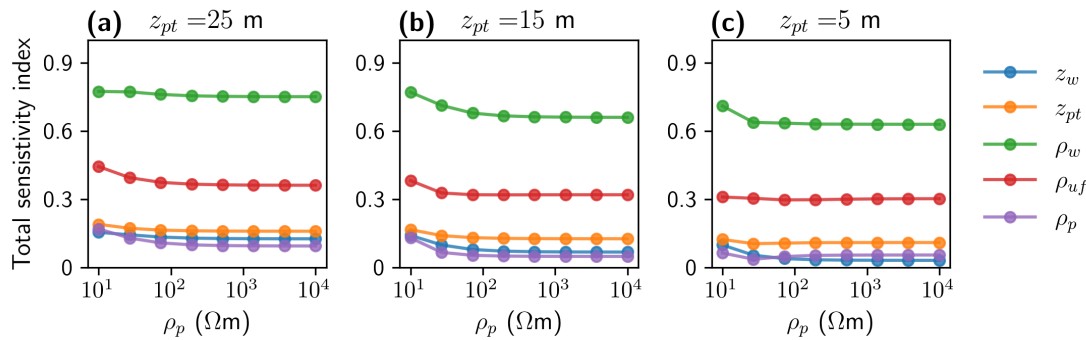

**Figure 6.** Global sensitivity results for the Bykovsky field site considering different 1D model scenarios with an IBPT at a depth of a) 25 m, b) 15 m, and c) 5 m.

We repeat these inversion runs considering different initial seeds of the random number generator (using different processors in parallel) until obtaining an ensemble $M_{F0}$ consisting of 416 models.



### 5.2.1 Ensemble analysis

After the inversion, we interpolate all models to a refined structured mesh before performing any posterior statistical analyses (Sect. 4.3). In Fig. 7a and b, we present the $\mu_{1/2}(\boldsymbol{M}_{F0})$ and IQR($\boldsymbol{M}_{F0}$) models calculated from the Drew Point model ensemble. The irregular variations in the IQR($\boldsymbol{M}_{F0}$) model and the bimodal distribution of $\rho_p$ (some models with $\rho_p < 500$ $\Omega$m and others with $\rho_p > 100,000$ $\Omega$m) already indicate different groups of models with distinct resistivity characteristics and IBPT positions.


In the next step, we performed cluster analysis (Sect. 4.3) and found that our ensemble $\boldsymbol{M}_{F0}$ can be divided into three model families ($\boldsymbol{M}_{F1}$, $\boldsymbol{M}_{F2}$, and $\boldsymbol{M}_{F3}$). In Fig. 7b-d and f-h, we present the $\mu_{1/2}(\boldsymbol{M}_{Fi})$ and IQR($\boldsymbol{M}_{Fi}$) models (where $i = 1, 2, 3$). Comparing these models illustrates that $\boldsymbol{M}_{F1}$ and $\boldsymbol{M}_{F2}$ present a similar IBPT shape dipping toward the open sea. However, for $\boldsymbol{M}_{F3}$ the IBPT position is dipping toward the coast which is not in agreement with our background knowledge of this field

site. When comparing the $\boldsymbol{M}_{F1}$ and $\boldsymbol{M}_{F2}$ models in more detail, we note that the IBPT position in $\boldsymbol{M}_{F1}$ is shallower than in $\boldsymbol{M}_{F2}$. Comparable to the Bykovsky example, models favoring high $\rho_p$ values tend to show increased depths of the IBPT resulting in thicker unfrozen sediments also near the coast. According to the depth of the IBPT and its gradients in profile direction (for $\boldsymbol{M}_{F1}$ and $\boldsymbol{M}_{F2}$), we laterally subdivide the model into four main parts. The first part is found at $x < 100$ m and it is characterized by an intermediate convex slope. The second part is found at $100 < x < 500$ m and the IBPT shows a gentle

convex slope whereas in the third part (at $500 < x < 700$ m) the IBPT is almost flat. Finally, the fourth part is found at $x > 750$ m, where the IBPT may be located at depths $\geq 20$ m.

We assess the fit performance for the residuals associated to the ensemble containing all models $\boldsymbol{\delta}_{F0}$, as well as for the three clustered model families $\boldsymbol{\delta}_{F1}$, $\boldsymbol{\delta}_{F2}$, and $\boldsymbol{\delta}_{F3}$ (Fig. 8). We calculate $\mu_{1/2}(\boldsymbol{\delta}_{Fi})$ and IQR($\boldsymbol{\delta}_{Fi}$) (where $i = 0, 1, 2, 3$) in a

pixel-wise fashion and present them as pseudosections in Fig. 8a-h. When comparing these pseudosections to each other, we notice that $\mu_{1/2}(\boldsymbol{\delta}_{Fi})$ indicate similar fits of the data in terms of amplitudes and pseudosection patterns (although with slightly higher values for $\boldsymbol{\delta}_{F3}$). When comparing the IQR($\boldsymbol{\delta}_{Fi}$) plots, we note that IQR($\boldsymbol{\delta}_{F0}$) is characterized by several patches which are less prominent in the clustered residuals Fig. 8f-h. This indicates that our clustering results are properly grouping models with similar residuals. Furthermore, we associate the vertical feature at $x = 400$ m in Fig. 8e-h to the variation in our models to

locate the left edge of a bulge structure of the seabed (see Fig. 1f). This illustrates the applicability of exploring such residual statistics to identify possible drawbacks in our inversion results and, thus, allow us to re-evaluate our parameterization strategy. For example, we might consider to improve the inversion results by adding a node (to our sums of arctangent functions) around $x = 400$ m. The overall statistics RMSLE($\boldsymbol{\delta}_{Fi}$), IQR($\boldsymbol{\delta}_{Fi}$), and $q_{90}(\boldsymbol{\delta}_{Fi})$ (where $i = 0, 1, 2, 3$) are presented as histograms in Fig. 8i-l. The histograms in Fig. 8i are characterized by bimodal distributions. Such bimodal distributions are less pronounced

for the clustered families (Fig. 8j-l), however, small tails to the right are also evident for $\boldsymbol{\delta}_{F1}$ and $\boldsymbol{\delta}_{F2}$. One may tend to reject the models falling in these tails, especially, when using the mean to estimate the central trend. However, because we consider robust statistical measures (e.g., median and IQR), we not expect a significant impact from these models on our results and



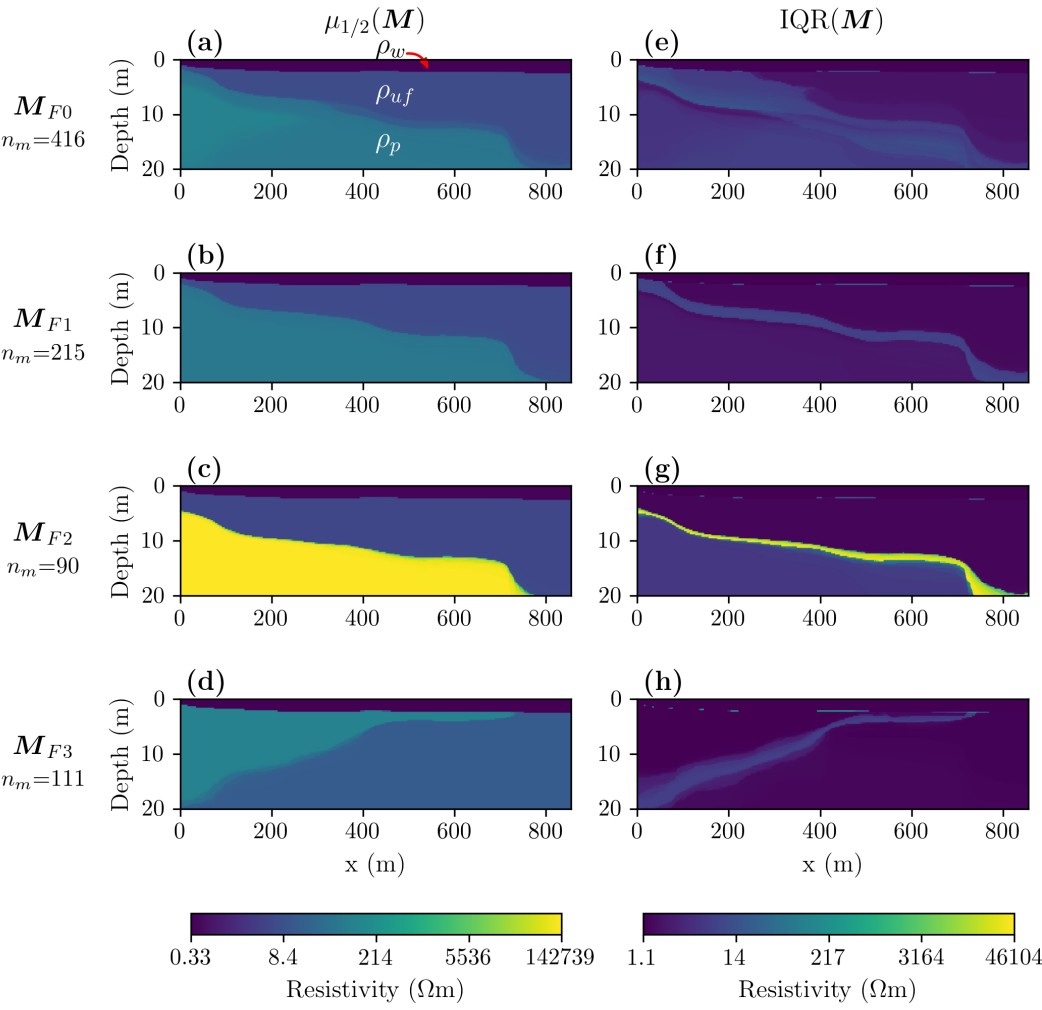

**Figure 7.** Inversion results for the Drew Point data set illustrated as summary statistics for all obtained models $M_{F0}$ and for three model families $M_{F1}$, $M_{F2}$, and $M_{F3}$ as found by cluster analysis. a)-d) Median and e)-h) interquartile range models. For each $M_{Fi}$, $n_m$ represents the number of models in the corresponding ensemble.

conclusions.

### 5.2.2 1D inversion of synthetic data

Following Sect. 4.4 and Sect. 5.1.2, we perform 1D inversions of a synthetic data set created considering a 1D subsurface model (see "Input model" in Table 3). The 1D model parameters were chosen by analyzing our 2D model solutions (e.g., Fig. 7b-c at

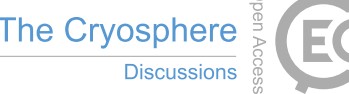

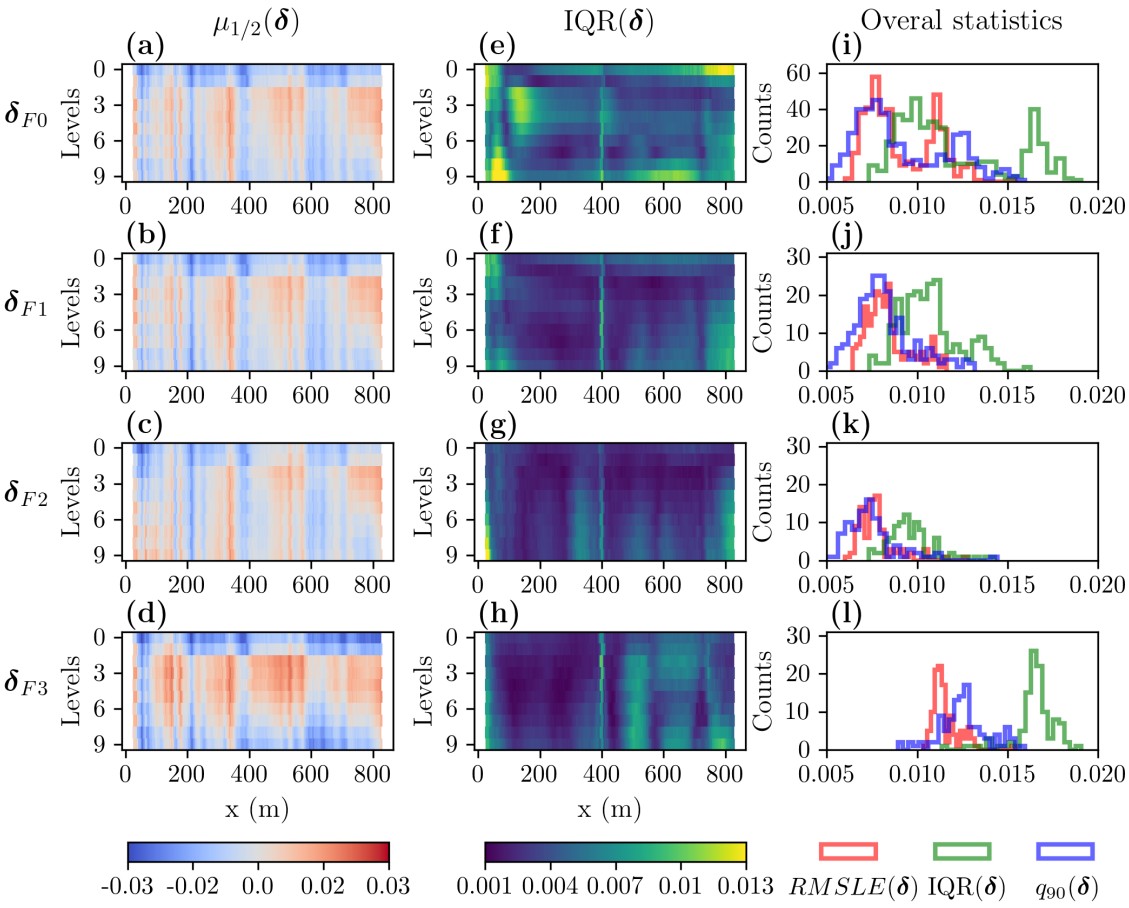

**Figure 8.** Summary statistics of the residuals for the Drew Point data set corresponding to all models $\boldsymbol{\delta}_{F0}$ and for the three clustered families $\boldsymbol{\delta}_{F1}$, $\boldsymbol{\delta}_{F2}$, and $\boldsymbol{\delta}_{F3}$. a)-d) Median and e)-h) interquartile range calculated in a pixel-wise fashion. i)-l) Histograms illustrating the overall distributions of different statistical measures including RMSLE($\boldsymbol{\delta}$), IQR($\boldsymbol{\delta}$), and $q_{90}(\boldsymbol{\delta})$.

$x \approx 600$ m). Note $\rho_p$ is the same as in the 1D synthetic example from Sect. 5.1.2 which allows us to better compare the results of our 1D synthetic exercises. We calculate the forward response of 10 quadripoles considering the same electrode configura-

tions as used for recording the Drew Point field data (Table 1). We invert the simulated apparent resistivity data considering two scenarios for constraining $z_w$, $\rho_w$, and $\rho_{uf}$, while the constraints for $z_{pt}$ and $\rho_p$ remain unchanged (see Table 3). The resulting inverted models are shown in Fig. 9a and c. For all models, we have achieved RMSLE $< 0.007$, which is equivalent to the noise level applied to the calculated synthetic data and comparable to the RMSLE achieved for the 2D inversion results of the Drew Point field data. Comparing the results shown in Fig. 9a and c illustrates that the applied constraints improve the

median model. However, we also observe an increase in the variability of the models around $z_{pt}$ in Fig. 9c. Additionally, from





all the models visualized in Fig. 9a and c, we calculate the corresponding posterior correlation matrix (Fig. 9b and d). For both cases, we see that the largest negative correlations are found for the model parameter pairs $[\rho_w, \rho_{uf}]$ and $[z_{pt}, \rho_w]$ while the most significant positive correlation is found for $[z_{pt}, \rho_{uf}]$ (note that the absolute correlations of these model parameter pairs are larger in Fig. 9d). Finally, we want to point out that the signs for the most significant parameter correlations are the same
as the ones found for Bykovsky in Fig. 4d.

**Table 3.** Parameters of the 1D synthetic model of Drew Point and for two scenarios indicating the lower and upper bounds parameter constraints.

|  | Input model | Scenario 1 | Scenario 2 |
|---|---|---|---|
| Depth seawater $z_w$ (m) | 2 | 1.5, 2.5 | 1.9, 2.1 |
| Depth IBPT $z_{pt}$ (m) | 12 | 3.5, 20 | 3.5, 20 |
| Resistivity seawater $\rho_w$ ($\Omega$m) | 0.4 | 0.2, 2 | 0.2, 0.6 |
| Resistivity unfrozen sediments $\rho_{uf}$ ($\Omega$m) | 5 | 0.2, 100 | 0.2, 20 |
| Resistivity permafrost $\rho_p$ ($\Omega$m) | 4,000 | 1, 200,000 | 1, 200,000 |

### 5.2.3 Sensitivity analysis

For sensitivity analysis, we consider the two model scenarios indicated in Fig. 10. In the first scenario, we consider the same input model as for the 1D inversion exercise (Table 3). In the second scenario, we set $z_{pt} = 16$ m while all other parameters
remain unchanged. From analyzing the cumulative sensitivity plots (Fig. 10a and d), we infer that an interpretation of our inversion results should focus on the area around the inner electrodes (i.e., if the boat is moving toward the right to collect additional sounding curves, our interpretation of the inverted model should start at $x \sim -10$ m). When analyzing Fig. 10b and e, we see that we are most sensitive to the water layer. Interestingly, when comparing Fig. 10c and f in detail, we realize that the sensitivity distribution in Fig. 10c reaches the IBPT interface while the sensitivity distribution in Fig. 10f is almost null for depths $> 12$ m.


We perform the global sensitivity analyses (Sect. 4.5) considering 1D models described by five model parameters as used for the above presented 1D inversions. We consider models where $z_w = 2$ m, $\rho_w = 0.4$ $\Omega$m, and $\rho_{uf} = 5$ $\Omega$m are fixed, while $\rho_p$ varies between 10 $\Omega$m and 10,000 $\Omega$m (eight values in total), and the IBPT is located at three different depths; i.e., $z_{pt} = 16$ m, $z_{pt} = 10$ m, and $z_{pt} = 4$ m (Fig. 11a-c). For the calculation of the total sensitivity for each of our five parameters in the
resulting 24 models, we set the parameter ranges to $z_w = [0.5, 3]$ m, $z_{pt} = [3.5, 20]$ m, $\rho_w = [0.2, 20]$ $\Omega$m, $\rho_{uf} = [1, 20]$ $\Omega$m, $\rho_p = [5, 20,000]$ $\Omega$m. For these specific models and parameters ranges, our results (Fig. 11) suggest that $\rho_w$ and $\rho_{uf}$ are the most influential parameters and the other three parameters ($z_{pt}$, $z_w$ and $\rho_p$) are characterized in all cases by rather low total sensitivities. Furthermore, we also notice in Fig. 11a-c that $\rho_w$ is the parameter showing the most significant changes when varying $\rho_p$ and $z_{pt}$.


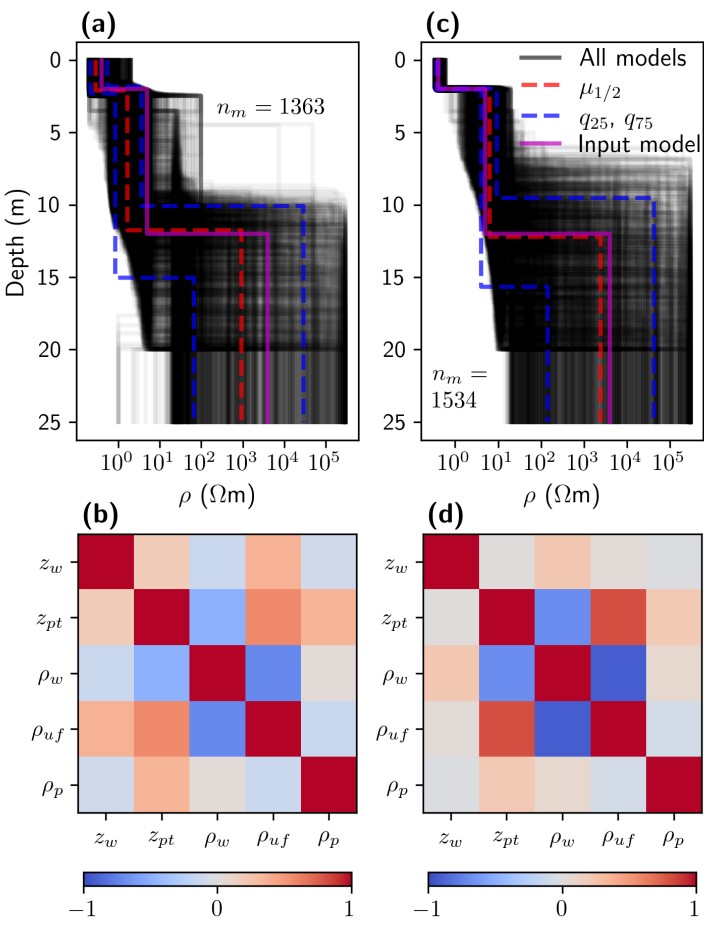

**Figure 9.** 1D inversion results of synthetic data for 1D subsurface scenarios developed for the Drew Point field site. a) Ensemble of solutions and b) the corresponding correlation matrix for scenario 1 (water layer parameters with large freedom during inversion), and c)-d) the same for scenario 2 (with constrained $z_w$, $\rho_w$, and $\rho_{uf}$). Note, black lines in a) and c) are plotted with transparency and, therefore, the darker areas indicate higher densities model. Each ensemble contains $n_m$ model solutions.

# 6 Discussion

Knowledge of how fast permafrost thaws would improve predictive models of greenhouse gas release and coastal erosion, as well as coastal infrastructure design. The ERT method has been successfully used to image the unfrozen sediments overlying the permafrost layer in subsea permafrost environments, especially using smooth inversion approaches (e.g., Overduin et al., 2012; Pedrazas et al., 2020). In typical subsea permafrost environments, there might be a gradual transition zone consisting of a mixture of water and ice between fully unfrozen and frozen ice-bonded sediments. However, during ERT inversion, the nature of this transition can be either enlarged when using smooth inversion approaches, or reduced to a single interface when

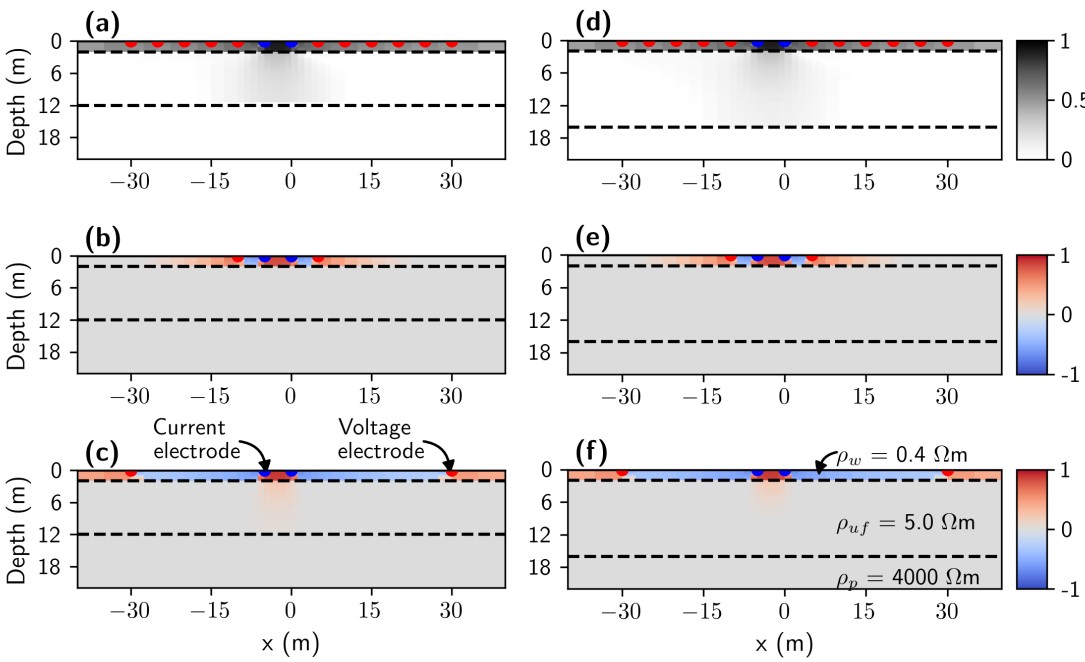

**Figure 10.** 2D normalized sensitivities for two different model scenarios developed for the Drew Point field site. Position of the the IBPT at a depth of a)-c) 12 m, and d)-f) 16 m. From top to bottom, we show the cumulative sensitivity and the sensitivity for the shortest and widest quadripole, respectively.

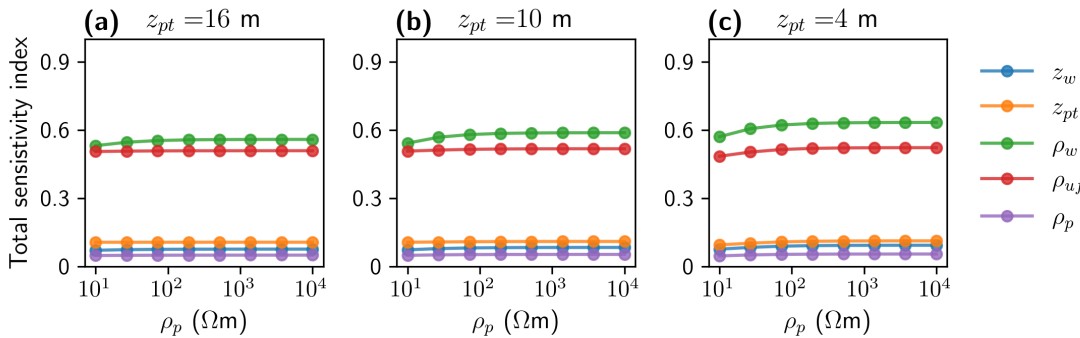

**Figure 11.** Global sensitivity results for the Drew Point field site considering different 1D model scenarios with an IBPT at a depth of a) 16 m, b) 10 m, and c) 4 m.

using layer-based strategies. Whether we have a smooth or sharp transition between frozen and unfrozen sediments, there must be a threshold in the ice content that creates sufficient contrast in resistivity also influencing the penetration of the injected current and, thus, our apparent resistivity measurements (e.g., Kang and Lee, 2015). In this study, we considered a layer-based





model parameterization to target (during ERT inversion) the interface defined by such resistivity contrast (interpreted here as the IBPT) including estimates of uncertainties.

## 6.1 Insights from our parameterization and inversion strategies

We used a 2D layer-based model parameterization to globally invert marine ERT data and obtain different ensembles (e.g., after cluster analysis) of model solutions. We demonstrated with the two case studies that such ensembles allow us to reliably

image the IBPT position with its associated uncertainties. The main advantage of using a layer-based model parameterization strategy is that we do not assume an arbitrary resistivity threshold or gradient to interpret the IBPT position as needed for the interpretation of smooth inversion results. This may be advantageous to compare ERT profiles collected at the same position in different years to track changes along the IBPT or in environments where the freezing point of the sediment porewater changes spatially. For example, offshore surveys that encounter submerged hypersaline lagoon deposits may show relatively low re-

sistivity values for partially frozen sediments compared to colder ice-bonded permafrost with fresh porewater (Angelopoulos et al., 2021). Indeed, this interface may be related to a threshold in ice content; however, associating the IBPT with a certain ice content requires calibration with direct sampling. Such thresholds may vary from site to site depending on properties of the sediments including temperature, grain size distribution, and the salinity of the porewater. Furthermore, we consider it convenient to use the sum of arctangent functions to parameterize the IBPT because there may be cases where the IBPT position varies

steeply (as the ones we identified at the end of the median models in Figs. 2 and 7 ) associated, for example, with submerged thermokarst structures or changes in the ratio of coastal erosion vs. degradation rate.

To find reliable and stable model solutions, we performed several experiments where we ran the PSO several times to find the appropriate parameter settings for the inversion. We noticed that constraining the seabed interface using the water depth

data as collected with an echo sounder (allowing for depth variations in the range $\pm 0.15$ m, which is the approximate error level of the echo sounder for water depths $< 5$ m) and the resistivity data as recorded with a CTD device (allowing variation between the minimum and maximum values of the measured data) significantly improved the inversion results and reduced the non-uniqueness of the inverse problem (as also demonstrated by the results of 1D synthetic inversion experiments). Although in this study we only analyzed the results considering a three-layer model parameterization (water and unfrozen sediments

overlying an ice-bearing permafrost layer), in the experimental phase, we also explored the possibility of using more than three layers. However, we did not observe any significant improvement in the final median models when increasing the number of layers and, thus, restricted our inversion and analyses to three-layer scenarios.

One disadvantage of using a layer-based model parameterization relying on homogeneous layers is that it is not possible

to resolve small-scale resistivity variations (e.g., horizontal heterogeneities at a spatial scale of meters). However, our workflow allowed us to inspect and evaluate model performance including the appropriateness of the model parameterization. For example, in the Bykovsky and Drew Point case studies, we observed in the residual pseudosections some regular lateral variations (see Figs. 3 and 8). This indicates that we were not completely explaining the data, either because of lateral subsurface





resistivity variations or 3D effects. To tackle this problem, it could be beneficial to measure 3D bathymetric data around each
ERT profile and collect additional (parallel and perpendicular) ERT profiles to better understand 3D resistivity variations at
our field sites. Furthermore, direct measurement of the resistivity of water and unfrozen sediments (e.g., using additional water
samples and drilling cores) might help to inform the model parameterization (e.g., account for lateral variations) and inver-
sion strategies. We should notice that adding complexity to our model parameterization comes with the trade-off of increased
computational cost to solve the inverse problems. An alternative to obtaining more complex resistivity models is to use our
global inversion results (considering homogeneous resistivity layers) as reference models to perform local inversions relying
on a grid-based model parameterization.

    The error level of ERT data is usually unknown; especially, for marine data, where repeated or reciprocal measurements are
not possible because the data are acquired while the boat is moving. This represents a challenge during the inversion when
specifying an appropriate fit level. One alternative to get insights into the noise level is to perform repeated measurements in
a static fashion (avoiding bending of the cable by wind or swells) for a certain section of the profile. For example, this can
be achieved at the coast on a calm day where one end of the cable is secured to the beach and the other end is fastened to
an anchored boat. However, such repeat measurements were not available for our field sites. Therefore, we set our stopping
criterion by considering a fixed number of iterations rather than using a minimum threshold in our objective function. With this
approach, we obtained model solutions characterized by different fit levels. For example, for the Bykovsky data, we found RM-
SLE values between 0.025 and 0.038 (Fig. 3g-i), while for our Drew Point data, we found RMSLE values between 0.007 and
0.016 (Fig. 8i-l). Although the RMSLE values for Drew Point are significantly smaller than for Bykovsky, we found a family
of models in the Drew Point study, which was considered as geologically unrealistic (Fig. 7d). This highlights the importance
of estimating different ensembles of solutions with different fit levels.


### 6.2   Parameter learning from 1D inversion

Our 2D inversion results showed large variations in the modeled resistivities of the permafrost, and we also noticed that, typ-
ically, the variabilities of IBPT position increase with depth. These observations indicate decreasing resolution capabilities of
our ERT data with depth and limited penetration of the injected current in the frozen permafrost layers. To better understand
these results in a more quantitative fashion, we reduced the number of parameters to five and performed selected 1D inversion
experiments using synthetic data inspired by our 2D inversion results. Because such 1D inversions are significantly faster than
2D inversions, they represent an efficient way to explore the influence of constraining different parameters. For example, we
noticed from our 1D inversion results that constraining the water layer significantly decreased the non-uniqueness of the in-
verse problem, which is essential, for example, for a reliable estimation of the IBPT position and for establishing petrophysical
relations (e.g., to estimate porewater salinity and ice content). Additionally, we noticed that the 1D inversion results for the
Bykovsky data (Fig. 4c) provided similar uncertainties around the IBPT as the 2D inversion results at $x = 150$ m (Fig. 2d-f ).
However, the 1D inversion results for the Drew Point data (Fig. 9c) showed uncertainties around the IBPT three times larger



compared to the 2D inversion results at $x \approx 600$ m (Fig. 7f and g). This indicates that there is no general best way of using the results of such complementary synthetic 1D studies; the success and feasibility rather depends on the characteristics of the field site and analyzed data set. On the other hand, we can use our 1D inversion results to assess the posterior correlation matrix that, as we showed in our examples, can be helpful to identify interactions between the models parameters. Furthermore, comparing the changes across different posterior correlation matrices (e.g., associated with different model constraints) can help us detect changes in the parameters interactions and, thus, quantify the impact of our model constraints. Such straightforward but informative analysis provides a deeper understanding of the inversion process and the suitability of the entire inversion strategy.

Our 1D inversion results indicated some problems if we want to infer relative permafrost characteristics from ERT measurements. The 1D input models for our 1D synthetic examples (see Table 2 and Table 3) assumed identical resistivities of the ice-bearing permafrost layer ($\rho_p = 4,000$ $\Omega$m) and similar resistivities for the unfrozen sediments (as found by our 2D inversion results). In contrast, the resistivity and depth of the seawater layer between both models were set according field measurements at our field sites. Although the resistivities of the unfrozen and frozen layers were similar in both models, we noticed that for model scenarios derived from the Bykovsky site, the inverted $\rho_p$ values were overestimated because $q_{25}$ and $q_{75}$ models indicate $\rho_p$ values larger than the input $\rho_p$ (Fig. 4c). On the contrary, for settings inspired by the Drew Point field site, the input $\rho_p$ fell within the range defined by $q_{25}$ and $q_{75}$ models but showed more significant variabilities than our 1D Bykovsky experiment (Fig. 9c). These results demonstrated the influence of the depth and resistivity of the seawater layer in the inverted models which may be critical for subsequent interpretations. For example, assuming the resistivity of the sediments increases with ice-content (e.g., Pearson et al., 1986; Fortier et al., 1994; Kang and Lee, 2015), such results may lead us to conclude that the ice-bearing permafrost layer holds higher ice content at Bykovsky compared to Drew Point (assuming the sediments have the same temperature and porewater salinity). Over or under-estimating the resistivity may lead to potentially erroneous interpretations, for example, related to the sediments ice content, temperature, and composition. We would need complementary field information or further analyses like sensitivity assessments to avoid misleading interpretations.

## 6.3 System understanding with sensitivity analysis

We obtained an additional model understanding (e.g., in view of delineating confident and reliable model areas) by performing sensitivity analyses. From our examples, we learned that if the resistivity of the seawater were higher than the resistivity of the unfrozen sediments (as in the Bykovsky case study, Fig. 5), this would result in increased sensitivities inside the unfrozen sediments and, thus, to changes along the IBPT position. This type of situation may be prevalent in subsea permafrost areas affected by freshwater river discharge in summer. On the other hand, we noticed that if the seawater were less resistive than the unfrozen sediments (e.g., as in the Drew Point case study, Fig. 10), we were more sensitive to the water layer and, therefore, to bathymetric changes. This emphasizes the importance of accurate water depth measurements. We highlight the fact that although the local 2D sensitivities for the Drew Point data were rather small for the unfrozen sediments, the IQR of the models (Fig. 7f-g) showed equivalent variability around the depth of IBPT (1.5 to 2 m for depths $\sim 12$ m) in comparison to the



Bykovsky example (Fig. 2d-f), where the sensitivities showed a more pronounced influence within the frozen sediments.

This study used global sensitivity analysis considering only five parameters as needed for our 1D inversion examples. The
Sobol approach proved to be a powerful method to distinguish the most influential parameters. After evaluating how the permafrost resistivity and the IBPT position may influence the rest of the parameters in our 1D three-layer examples, we noted some relevant differences. For example, in the Bykovsky example (Fig. 6), we noticed that for larger values of $\rho_p$ and shallower $z_{pt}$ the total influence of the rest of the parameters decreased. On the other hand, for the Drew Point example (Fig. 9), increasing $\rho_p$ increased the total sensitivity of the rest of the parameters, while varying $z_{pt}$ at shallower depths mainly increased
the influence of $\rho_w$. We also want to highlight that $\rho_w$, $\rho_{uf}$, and $z_{pt}$ (which were the parameters with the largest total sensitivity in both examples) were also the parameters that formed model parameter pairs showing the largest correlation (see Fig. 4d and Fig. 9d). Encouragingly, $\rho_w$ and $\rho_{uf}$ can be informed from CTD casts and shallow sediment sampling, respectively. We must be aware that such a global sensitivity analysis is highly dependent on the pre-defined constraining parameter range and should be applied to address specific questions to allow, for example, parameter reduction or to guide our sampling strategies and
experimental design.

### 6.4 Subsea permafrost features (Bykovsky vs. Drew Point)

The inverted ERT profiles yielded new insights about how subsea permafrost thaws, because the Bykovsky Peninsula (Bykovsky) and Drew Point are characterized by distinct seawater properties and geological histories. The Bykovsky 2D inversion results at $x = 150$ m, which corresponds to an inundation period of 357 years assuming an erosion rate of 0.42 m per year (e.g.,
Lantuit et al., 2011), showed a median depth to the IBPT of $\sim 15$ m (Fig. 2). This resulted in an average degradation rate of $\sim 0.04$ m per year. On the other hand, the Drew Point 2D inversion results at $x \approx 600$ m showed a median depth to the IBPT of $\sim 12$ m. Note that this location coincides with the 1955 coastline position (see Fig. 1e-f), which corresponds to 63 years of inundation yielding an average degradation rate of $\sim 0.19$ m per year. At Bykovsky, 63 years of inundation (again assuming an erosion rate of 0.42 m per year) corresponds to an offshore distance of $\sim 26$ m which corresponds to a median IBPT depth
in the 2D inversion results (Fig. 2) at most 6 m. Although the mean annual IBPT degradation rate slows with inundation time as the temperature gradient driving diffusive heat fluxes weakens (Angelopoulos et al., 2019), it is evident that the permafrost at Drew Point may thaw faster, presumably because Drew Point sediments are primed with salts in the pore space prior to inundation (Black, 1964; Sellmann, 1989).

Since salt diffusion is typically slower than heat diffusion (Harrison and Osterkamp, 1978), the IBPT degradation rate at Bykovsky should theoretically be faster than at Drew Point, provided that the permafrost sediments are similar. However, it appears that dissolved salts in the pore space of the sediments at Drew Point play an important role in lowering the permafrost freezing point and resulting in higher IBPT degradation rates than at Bykovsky. In fact, the top of onshore cryotic and saline unfrozen sediment layers (cryopegs) were observed near the Drew Point shoreline during coring (Bull et al., 2020; Bristol
et al., 2021). This can lead to interpreting a faster IBPT degradation rate at Drew Point compared to Bykovsky in two ways:



1) a layer of submerged Drew Point sediments was already unfrozen upon inundation (e.g., $M_{F2}$ in Fig. 7); 2) the frozen layers at Drew Point contained less ice and had a lower freezing point. Jones et al. (2018) suggested that warming permafrost temperatures at Drew Point (3 to 4 °C over the past several decades) have made saline permafrost more susceptible to erosion, potentially contributing to the enhanced coastal erosion rate (2.5 times that of the historical average) observed between 2007

and 2016. Warming by seawater submergence would presumably result in even larger cryopeg spreading and IBPT degradation.

As shown in Fig. 1a and e, the coastal plains at our field sites consist of numerous thermokarst lakes and drained lake basins. When thermokarst lakes are breached by coastal erosion, the unfrozen sediments underneath the lake become integrated into the subsea permafrost environment, leading to U-shaped electrical resistivity structures. For example, Angelopoulos et al.

(2021) showed steep IBPT gradients along ERT profiles parallel to the southern Bykovsky shoreline that traverse submerged thermokarst and undisturbed permafrost. These authors also suggested that drained lake basins, which have undergone thaw-refreeze cycles, are more susceptible to quicker thaw compared to undisturbed terrain. Comparing the first 400 m of our inverted median models for our field sites, we noticed that, in general, the IBPT at Drew Point is smoother than at Bykovsky. This might be the result of the higher erosion rates at Drew Point ($> 10$ m per year) than in Bykovsky ($< 1$ m per year) that

expose coastal areas to inundation in a shorter time. Because of the longer inundation time at Bykovsky, we expect fluctuations in different environmental controls (e.g., water temperature, seawater salinity) that might result in step-like features as the one at $x \approx 280$ m. Furthermore, layered strata alternating between ice-rich and relatively ice-poor sediment may also contribute to step-like IBPT features. Similarly, in the Drew Point 2D inversion (Fig. 7a-c), there was a steep median IBPT decline observed at $x \approx 750$ m where the IBPT deepens from $\sim 12$ m to $\geq 20$ m. Although the resolution capabilities of our ERT data at these

depths are limited, we suggest that thermokarst processes prior to seawater submergence may be responsible for the nature of this IBPT dip.

## 7 Conclusions

In this study, we illustrated how we could use ERT data to reliably estimate the IBPT position in shallow coastal areas of the Arctic. We found that using a layer-based model parameterization helps us target the IBPT position directly from the

inversion of ERT data with the trade-off of omitting small-scale heterogeneities. To improve the inversion result, we noticed that constraining the water layer depth and resistivity reduces the non-uniqueness of the ERT inverse problem improving the estimation of the resistivity of the unfrozen sediments (talik and/or cryopeg) and the IBPT position. However, even constraining the water layer, we still found large variabilities in the resistivity of the frozen sediments. We suggest that constraining the resistivity of the unfrozen sediments (e.g., sediment sampling) during ERT inversion could improve resistivity estimates of

the frozen layer and, thus, further permafrost's physical properties (e.g., ice content). Properly imaging the IBPT position may allow us to improve the estimation of the permafrost degradation rate, which might be used to better understand greenhouse gas emissions and coastal erosion processes. The workflow and methods presented in this study can guide future field campaigns and may be used as a reference for more detailed parameterizations and/or inversion strategies.





*Data availability.* The data presented in this paper for our Bykovsky field site are available from the PANGAEA website (https://doi.org/10.1594/
PANGAEA.895887).

*Author contributions.* Mauricio Arboleda-Zapata: conceptualization, formal analysis, methodology, software, visualization and writing.

Michael Angelopoulos: conceptualization, data collection, project administration, supervision, review and editing.

Pier Paul Overduin: conceptualization, project administration, supervision, review and editing.

Guido Grosse: funding acquisition, data collection, project administration, review and editing.

Benjamin M. Jones: project administration, review and editing.

Jens Tronicke: conceptualization, formal analysis, methodology, supervision, funding acquisition, review and editing.

*Competing interests.* The authors declare that they have no known competing financial interests or personal relationships that could have appeared to influence the work reported in this paper.

*Acknowledgements.* Mauricio Arboleda-Zapata was supported by grant 57395813 from the German Academic Exchange Service (DAAD)
in the framework of the Research Training Group "Natural hazards and Risks in a Changing World" NatRiskChange 2043/2 funded by the Deutsche Forschungsgemeinschaft (DFG). Benjamin M. Jones was supported by grant OISE-1927553 from the US National Science Foundation and Paul P Overduin by the EU Nunataryuk project. Fieldwork has been supported by the European Research Council project PETA-CARB (ERC-338335) to Guido Grosse and AWI base funds. We thank Josefine Lenz and Juliane Wolter for their help during fieldwork at Drew Point and Mikhail N. Grigoriev (Melnikov Permafrost Institute, Yakutsk, Russia), who helped to realize the entire Bykovsky study
program. Finally, our sincere thanks to the floatplane pilot Jim Webster for his many years of service supporting fieldwork in Alaska.



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
