# Peer review of "Exploring the capabilities of electrical resistivity tomography to study subsea permafrost"

_The Cryosphere, 2022_

## Author Comment (AC1)

[Figure]

Figure 1. Smooth inversion result for our Bykovsky data set constraining the water layer.

[Figure]

Figure 2. Smooth inversion result for our Drew Poin data set constraining the water layer.

---

## Author Comment (AC2)

[Figure]

Figure 1. CTD profiles taken along our ERT transect at our Drew Point field site. Note that the line colors indicate the distance from the shoreline.

---

## Author Response (AR1)

**Answer comments Reviewer 1**

*Line 29: "layer (or body)" -> "layer or body". There are many spots throughout the paper where parentheses are unnecessary and are actually a bit distracting, as they disrupt the flow of the sentence. Revising throughout the paper will improve the flow and make your ideas easier to understand.*

Following this recommendation, we removed some parentheses to allow for a more fluid read.

*Line 66: "the less conductive is the medium" -> "the more resistive the medium", since you've been using resistivity to describe the water and sediments throughout the paragraph.*

We agree. For consistency, we replaced in the whole text the word conductivity by resistivity.

*Line 86: The phrasing of this sentence makes it sound like Arboleda-Zapata et al. (2022) also looked at the IBPT. I think it makes more sense to omit "also around the IBPT" to avoid this confusion.*

We agree with this comment, and we rephrased the corresponding sentence.

*Figure 1 caption, line 3: "read line" -> "red line"*

We fixed this misspelling error.

*Figure 1 (b) and (f): I would use a different color besides red to indicate historical coastlines (since you've already indicated the red lines show the ERT profiles). Maybe a black dashed line to agree with Figure 1e would be better.*

We followed this recommendation and changed the color of the red line in Fig. 1b and f to black.

*Line 159: I disagree – I don't find these plots particularly useful and would omit them in the final paper. Even with your interpretation of higher noise levels in levels 7 and 8 in the Bykovsky dataset, I think this is easier to see in Figure 1c than it is in 1d (and would argue that this is better described as variability than noise, because it may be caused by real features).*

We agree with this suggestion and removed the corresponding plots. Additionally, we updated the corresponding text and sentences.

*Line 181: It would be nice to specify that these are features you might expect to see at your study sites. Maybe something like "Allowing for abrupt changes is important in permafrost environments where high structural variability is often found. At our sites, we could expect to see sharp boundaries due to…"*

We rewrote the corresponding sentence in the second paragraph of Sect 4.1 and now it is highlighted that some of these structures might be present at our field site.

*Line 202: So every mesh is different? How is the mesh structure determined? More explanation is needed here.*

Following this comment, we extended this in the text and added two references (Schewchuk, 1996 and Arboleda-Zapata et al., 2020) where this strategy is further discussed.

*Line 232: "we not" -> "we do not"*

We updated this accordingly.

*Line 264: It's not clear to me what "considering five nodes for each interface" is referring to. Does this mean that each interface is parameterized by five depths along the survey line? Clarification would be helpful here.*

Following this comment, we extended our parameterization strategy in section 4.1 and added the arctan function. Additionally, Because we considered the same number of nodes and interfaces for both of our case studies, to avoid repetition, we added this information in section 4.1 instead of in each case study.

*Line 284: This phrasing could be interpreted as a general observation that more resistive permafrost = deeper boundary. I think it's important to specify two things: 1) that this is specific to your model, not a general observation, and 2) that this is due to a model equivalence/non-uniqueness problem (which will also help to introduce the following section).*

We agree with this suggestion and reformulated this sentence.

*Figure 2: It would be nice if you showed the smooth inversion here as well, as it would provide a nice comparison for the layer-based models. Same comment for Figure 7.*

Comparing different inversion strategies is beyond the scope of this study. However, for completeness and as a reference base model, we added the smooth inversion results (see figures 1 and 2 in the attached file) to an appendix but without analyzing or discussing them in detail. The interested reader is referred to Angelopoulos et al.(2019), where the Bykovsky data set was inverted using a smooth inversion approach, and the obtained results have been discussed in detail.

*Line 318: "because" -> "and". This statement is more of an observation than an explanation. Same comment for line 526.*

We agree with this comment and reformulated this statement to illustrate how far we are from the input model.

*Line 351: Here, you could explicitly state that the low sensitivity to permafrost resistivity causes the error in your 1D models and contributes to the uncertainty in your 2D models.*

We agree with this comment and have added a corresponding statement.

*Figures 4 and 9: This is mostly personal preference, but I would find the correlation matrices easier to read if they only showed the lower left portion and omitted duplicate cells. I also find it difficult to estimate the magnitude of the correlations using the color scale alone and suggest printing the numerical values on each cell in addition to the color.*

We added the correlation values in the upper half of the correlation matrix. However, we prefer to show the entire correlation matrices also to highlight this symmetric property of the matrices.

*Line 487: "especially, for marine data," -> "especially for marine data"*

We fixed this misspelling.

*Line 499: You could also note that this highlights the importance of having an accurate estimate of data noise. Since the misfits for the model in Figure 7d were higher, this set of models could potentially be ruled out if they were found to exceed expected error levels.*

Following this comment, we also highlighted this in the corresponding statement.

*Line 533: "resistivity" -> "ice-bearing permafrost resistivity"*

We agree and have added the missing part as suggested by this reviewer.

*Line 615: It's great that the data are available. If possible, you could share your code as well so that others can easily reproduce and build on your work.*

The data used for our Bykovsky example is already available. We will also upload the Drew Point data set to the Pangea repository. At some point, we want to share the code once it is better organized and adequately documented. We highlight that there are already several available implementations of PSO. For example, in python, you will find an implementation under https://pyswarms.readthedocs.io/en/latest/index.html. Additionally, all the mesh manipulation and the forward solver were done in the freely available Python library pyGIMLi https://www.pygimli.org/. Our implementation consists of adding the interfaces with the arctangent function while preserving minimum mesh quality requirements.  A similar approach is also given by Akça et al.(2010).

**Answer comments Reviewer 2**

*Reading over the paper, I think what is missing is a thorough comparison to conventional, i.e. smoothness-constraint, ERT inversion. Since there is no ground-truth available, the authors cannot show that their approach provides superior accuracy in determining the IBPT. So the reader is somewhat left wondering why this additional computational effort is actually needed. Couldn't you achieve similar results by using "standard" processing schemes? To address this, I would suggest adding the results of a smoothness-constraint inversion to Fig. 2 and 7, which I believe will show the benefit of your inversion method clearly, and will highlight that the additional computational effort yields a more robust recovery of the subsurface structure.*

we added the smooth inversion in an appendix. Please note that a smooth inversion for the Bykovsky data set is already published and discussed in detail by Angelopoulos et al. (2019). As also indicated in our reply to a similar comment from reviewer 1, our aim is not to compare and evaluate different inversion approaches. We rather want to present an alternative approach to image the IBPT interface and estimate uncertainties in depth and resistivity when we do not have any borehole data, which is typical in subsea permafrost studies.

*Another more fundamental comment refers to the spatial heterogeneity of the water resistivity you are trying to image. You describe the two field sites as places with different flow patterns feeding freshwater into the coastal system. I believe that this is likely causing spatial heterogeneity in the water resistivity going from the coast further into the sea. Yet, in your inversion approach, you only address the variation in the thickness of the sea-water layer, but not its resistivity. Why are you not addressing this? Is it because the variability in rho_w is small enough that it does not affect the inversion (if so, can you show that?), or is there another reason for not addressing it?*

Our decision to use homogeneous resistivity for the water layer for both of our case studies is based on different CTD measurements near our ERT profiles. For the Bykovsky field site, CTD measurements offshore of the Bykovsky Peninsula in July 2017 (freely available at https://doi.org/10.1594/PANGAEA.895887) demonstrate that there is little vertical variation in the electrical resistivity (or conductivity) in the water column. Additionally, for a particular day (e.g., 29 July), there was up ~1 ohm-m of water resistivity variation laterally. Because we do not expect significant stratification, assuming homogeneous resistivity is appropriated. Although the CTD measurements indicate resistivity values around 13.7 ohm-m, we still allow resistivity variations between 11 and 15 ohm-m. We added this reasoning in first paragraph of sections 4.2.

For our Drew Point field site, CTD control points along our ERT transect indicate minimal lateral and vertical variation in water resistivity. To illustrate this, we added a supplement figure in our previous reply with the CTD cast profile at different offshore distances. As noticed in that figure, water resistivity is in the order of 0.42 and 0.44 ohm-m for the first 600 m with minor variations in the vertical direction. Because we have rather small resistivity variations both horizontally and vertically, we can justify our decision of using a homogeneous resistivity for the water layer in our Drew Point example. Although the CTD measurements indicate resistivity values around 0.43 ohm-m, we still allow resistivity variations between 0.2 and 2 ohm-m. We added this reasoning in first paragraph of sections 4.2.

*Line 38: Although I generally agree, you may want to check out the work by Wagner et al., who show an approach to get quantitative values of ice content from joint inversion of ERT and seismic data.*

We included this reference in the discussion to support discussion about ice-content estimation.

*Line 66: It might be better to stick with resistivity here rather than changing to conductivity.*

We agree. For consistency, we replaced the word conductivity with resistivity in the whole text.

*Figure 1 (d) & (h): What do you mean by sounding number here? Does this refer to the measurements per level? Is there really a need for this last panel? In the text you only refer to this plot to highlight the higher noise level, but I think you can also that comparing (c) and (g).*

Following common terminology, a sounding refers to a set of electrode configurations collected with different spacings around one central sounding location. In our studies, the used streamer had 10 channels and allowed us to measure ten electrode configurations with different spacings. As also suggested by reviewer 1, these plots have been removed for the new version of the manuscript.

*Line 160-161: Judging from c, it looks like levels 6 to 9 in general are noisier than the shallower ones.*

We rephrased these lines because Fig. 1d and h were removed. We also indicated that the data starts being noiser after level 6.

*Line 188: To improve clarity, it might be worth adding here how you describe the geometry of the interface. Are you using a specific function with x numbers of parameters, or do you have a layer thickness for each sounding location?*

Following this comment and a similar comment from reviewer 1, we added more details about our model and interface parameterization to section 4.1, including also the corresponding arctan function.

*Line 264: Is this an arbitrary number for the number of points of the interface, or where does it come from?*

This is now clarified in section 4.1

*Line 265: I'm not entirely sure I follow how you get to 36? Five nodes for two interfaces should be 10 parameters describing the thickness, and then you need a resistivity for the water column, unfrozen sediments and frozen sediments.*

This is now clarified in section 4.1

*Line 330: This argumentation is a bit weak. Only because you have some sensitivity does not necessarily mean that you can resolve subsurface structures and that you can interpret the inverted models.*

Having some sensitivity means that a change in resistivity may impact our cost function, thus, influencing the finally found resistivity model. As we already pointed out in line 332, a more conservative way may be to start our interpretation at a position of x = -25 m where most of the sensitivity is concentrated. However, while we agree that there is less sensitivity below the outermost electrodes close to the shoreline, our interpretation of features in the ERT inverted profiles was not based on the geophysical data alone, as discussed in lines 578 - 582. For example, the cryopeg features were encountered by drilling observations reported in Bull et al. (2020) and Bristol et al. (2021). Therefore, the interpretation of the nearshore features shown in MF2 (Figure 7) is plausible.  Because our answers are already stated in the original manuscript, we did not extend the statement in line 330.

*Line 335-336: These areas seem a little suspicious to me. Why do you first get almost no sensitivity, and then a comparable high value. This does not seem to agree with the expected sensitivity pattern.*

We agree that these sensitivity patterns may look odd for conventional sensitivity analysis assuming lower resistivity contrast. However, these sensitivity patterns can be obtained with such high resistivity contrast between the horizontal layers, as also pointed out by Spitzer (1998). To assess up to what point these sensitivities below the outer electrode are present, we calculated the sensitivity considering other depths to IBPT. We found that the sensitivity below the external electrodes is almost zero by setting a depth to IBPT of 20 m. Because this additional analysis does not impact our presented results, we left the statement in lines 335-336 as it is presented.

*Line 359: Why did you choose different PSO parameters for the two different sites? To compare the results, wouldn't it be better to use the same set of parameters?*

Generally, there is not a unique recipe to carry out PSO optimization. As a rule of thumb, setting the number of particles one to three times the number of parameters is a good compromise. Because we noticed that the inversion of the Drew Point data set was converging much faster than Bykovsky, we decided to lower the number of particles and the number of iterations to save some computational cost.

*5.2.3 Sensitivity analysis: Having a sensitivity study for each site feels a bit repetitive. Perhaps merging the two sensitivity studies would make sense?*

Although we recognize that it may sound repetitive, we found it appropriate to let it in the current positions. We wrote the sections in parallel, following the same structure and workflow. Some parameters considered in our sensitivity analysis are derived from previous subsections. We think that the current positions of the sensitivity sections are still appropriate for this manuscript, especially because the two subsea permafrost environments are so different.

*Line 464 - 468: I may have missed that, but where do you show that in the 2D case. As I understand, you invert for rho_w and z_w only.*

With this comment, we realized we did not mention the constraints for our 2D inversion in our methodology section. Instead they were added in the discussion section. We moved the constraints information from Discussion to section 4.2 where we think it is more appropriate to understand the subsequent sections.

**Other changes**

We have not changed important things, but we have corrected several typos and reworded some sentences for clarity.